# Scars in the Abyss: Reconstructing sequence, location and temporal change of the 78 plough tracks of the 1989 DISCOL deep sea disturbance experiment in the Peru Basin

Gausepohl, Florian[1,2], Hennke, Anne[1*], Schoening, Timm[1], Köser, Kevin[1],Greinert, Jens[1,2]

[1] GEOMAR Helmholtz Centre for Ocean Research Kiel, Kiel, Germany

[2] Christian-Albrechts-University Kiel, Kiel, Germany

*Correspondence to:* Anne Hennke (ahennke@geomar.de)

**Abstract**

High-resolution optical and hydroacoustic seafloor data acquired in 2015 enabled the reconstruction and exact localization of disturbance tracks of a past deep sea re-colonization experiment (DISCOL) that was conducted in 1989 in the Peru Basin during a German environmental impact study associated with manganese nodule mining. Based on this information, the disturbance level of the experiment regarding the direct plough-impact and distribution and re-deposition of sediment from the evolving sediment plume was assessed qualitatively. The compilation of all available optical and acoustic data sets available from the DISCOL Experimental Area (DEA) and the derived accurate positions of the different plough marks facilitate the analysis of the sedimentary evolution over the last 26 years for a sub-set of the 78 disturbance tracks. The results highlight the remarkable difference between natural sedimentation in the deep-sea and sedimentation of a resettled sediment plume; most of the blanketing of the plough tracks happened through the resettling of plume sediment from later created plough tracks. Generally sediment plumes are seen as one of the important impacts associated with potential Mn-nodule mining.

For enabling a better evaluation and interpretation of particularly geochemical and microbiological data a relative age sequence of single plough marks and groups of them was derived and is presented here. This is important as the thickness of resettled sediment differs distinctly between earlier and later created plough marks.

Problems in data processing became eminent for data from the late 80s, at a time when GPS was just invented and underwater navigation was in an infant stage. However, even today the uncertainties of underwater navigation need to be considered if a variety of acoustical and optical sensors with different resolution should be merged to correlate accurately with the absolute geographic position. In this study, the ship-based bathymetric map was used as absolute geographic reference layer and a workflow was applied for geo-referencing all the other datasets of the DISCOL Experimental Area until end of 2015. New high resolution field data were mainly acquired with sensors attached to GEOMARs AUV Abyss and the 0.5 x 1° EM122 multibeam system of RV SONNE during cruise SO242 -1. Legacy data from the 1980s and 1990s first needed to be found and compiled before they could be digitized and properly georeferenced for our joined analyses.

## 1 Introduction

### 1.1 Ecological risks associated with Mn-nodule mining from the deep seafloor

For several years, mining of manganese (Mn) nodules from the deep seafloor is again considered a worthwhile option to meet future resource demands. Several nations secure exploration contracts in areas beyond any national jurisdiction as they seek economic benefits and/or aim for technological leadership in terms of deep sea mining. Current plans for a future mining scenario involve collectors that will move on the seafloor gathering Mn-nodules from the top 10 to 30 cm of the sediment, most likely using a hydraulic collection mechanism (Kuhn et al., 2011, Oebius et al., 2001). This principle implies considerable consequences for the benthic environment in the mined area. Besides the removal of the Mn nodules as an important hard-substrate habitat on the abyssal plains (Purser et al., 2016; Vanreusel et al., 2016, Thiel et al., 1993), the mining activities will completely re-work the top sediment layers and re-suspend large amounts of sediment into the water column. Depending on the plume properties such as particle size, flocculation behavior, sediment mass per liter, and the prevailing current conditions these sediment particles might be transported outside the mined area. The deposition of this material will cause a secondary impact on the environment by clogging of filter feeders and burial of the sessile fauna, which are both adapted to the low sedimentation rates in the deep sea (Thiel and Schriever, 1989). Re-sedimentation of this material can also lead to differences in local geochemical gradients and consequently might influence the recolonization processes of the primary and secondary disturbed areas. To evaluate these effects on the environment, several benthic impact experiments (BIE) and one Recolonization Experiment, the German Research Project "Disturbance and Recolonization Experiment-DISCOL" (http://www.discol.de), have been conducted in the past within different large Mn-nodule areas, including the Peru Basin (Thiel and Schriever, 1989), the Central Equatorial Pacific (e.g. Burns, 1980; Fukushima, 1995) or the Indian Ocean Basin (Desa, 1997). Information about the sediment plume dispersal during the different large-scale disturbances are compiled in section 1.2. A review of the biological responses to such BIEs was recently presented by Jones et al. (2017) and studies by Simon-Lledó et al. (2019) in the DISCOL Experimental Area (DEA) show that colonization pattern-differences still exist between the disturbed and undisturbed areas even after 26 years

### 1.2 Summary of plume dispersal results of past benthic impact experiments (BIE's)

In the late 1970's, the first so called 'mining test' operations were conducted in the central North Pacific as part of the DOMES project (Ozturgut et al. 1978, 1980) that used a Suction dredge towed on skis to create a disturbance for illustrating potential mining impacts. Here, the experimental area was surveyed before, during and after the experiment, with each disturbance lasting for several hours (see Table A1 for details on location, duration, monitoring techniques and impacted area). For the first three tests in spring 1978, operated by Ocean Mining Inc. (OMI), detailed data about the induced sediment plume were derived from different sampling methods including sediment coring and sediment traps (see Burns, 1980, and details in Table A1) and results indicate a plume dispersal of up to 16 km downstream of the created disturbance (Table A1). Model results based on the OMI experiment indicate a sediment blanketing thickness of ≤1 mm beyond 400 m distance to single disturbance tracks. An extrapolation of these results for a potential mining scenario was performed and predicted a distribution of re-suspended sediment particles of up to 160 km distance (Lavelle et al., 1981).

Another 'mining test' phase in November 1978 focused on the distribution of a surface discharge plume (Ozturgut et al., 1980). During an 18 hour lasting operation by Deepsea Ventures Inc. / Ocean Mining Associates (OMA). A second seafloor mining test was conducted in November 1978 by the Ocean Mineral Company (OMCO) using a Remote Controlled self-propelled Miner (RCM) (Chung, 2009). This vehicle removed approximately 4 cm of the upper sediment layer (Khripounoff et al., 2006) creating a track of 1.5 m width (Miljutin et al., 2011). The aim of this experiment was mainly to test the mining technology and not to monitor the benthic impact of the plume. Hence, detailed information regarding the sediment plume dispersal right after the impact is missing. In 2004, the disturbed area was revisited and investigated for its ecological recovery (Mahatma, 2009; Miljutin et al., 2011) indicating only a near-track influence of re-deposited sediment.

Chronologically the next and largest ever created disturbance was conducted in the DISCOL Experimental Area in the Peru Basin. For creating the disturbance, a plough-harrow (8 m width) was towed 78 times crisscrossing through a circular area of 2 nautical miles in diameter (Thiel and Schriever, 1989). Due to technical problems the deployed nephelometers at that time did not detect the sediment plume and the amount of suspended material remains largely uncertain. Nevertheless, the presence of a plume in the water column about 6 hours after the last plough deployment was confirmed by visual observations (Thiel and Schriever, 1989). Numerical modeling predicted a dispersal of the suspended sediment for several kilometers with coverages of resettled material of >100 $gm^{-2}$ up to a distance of 2 km (Jankowski et al., 1996; Table A1). The effects of the disturbance were investigated just after the experiment (RV SONNE cruise SO61, Thiel and Schriever, 1989) as well as 0.5 (cruise SO64, Schriever, 1990), 3 (cruise SO77, Schriever and Thiel, 1992), 7 (cruise SO106; Schriever et al., 1996) and finally 26 years later (cruise SO 242; Boetius, 2015; Greinert, 2015) to document the environmental impact, the recolonization and sediment geochemical equilibration of the disturbed sites in comparison to a number of undisturbed reference sites in the vicinity.

Again north of the equator, the first large-scale benthic disturbance experiment in the eastern Clarion-Clipperton Fracture Zone (CCFZ) conducted by the United States was the Benthic Impact Experiment II (BIE-II) in 1993, using the "Deep Sea Sediment Resuspension System" (DSSRS) (Brockett and Richards, 1994; Tsurusaki, 1997) as disturbance tool (Trueblood and Ozturgut, 1997). The initiated sediment plume was monitored with camera systems, sediment traps and transmissiometers, which were moored in different distances from the tow zone in order to estimate the distribution areas of re-settled sediment and the plume dispersal in the water column. The studies revealed an area of strong sediment blanketing within the first 50 m downstream of the disturbance and a decreasing blanketing thicknesses with increasing distance. Moorings located 400 m away still detected suspended material passing by and also deployed sediment trap samples indicated a maximum "blanketing" thickness of 1 mm. In contrast to these data camera observations suggested a sediment blanketing thickness of 1– 2 cm close to the disturbance zone (Jones, 2000) already indicating that the sediment traps might have missed the additional sediment transport of initiated gravity flows just above the seafloor.

One year after the American experiment, the Metal Mining Agency of Japan (MMAJ) carried out another disturbance study within the CCFZ, the "Japan Deep sea Impact Experiment" (JET) in 1994 (Fukushima, 1995). The disturbance was again created with the DSSRS (Tsurusaki, 1997). The distribution of the initiated sediment plume was analyzed using two different approaches. One approach measured the thickness of the blanketing sediment layer using sediment traps and spatially interpolated the results using Kriging. A dispersal of 2.5km in

length and approximately 1km in width was calculated and a maximum blanketing thickness of 2.6 mm was determined (Barnett and Suzuki, 1997). The second approach used visual data from deep-towed camera surveys to estimate the extent of the sediment blanketing that covered the Mn-nodules. Respective results show that the 'heavy' re-sedimentation area, defined by a thickness > 0.26 mm did not extend for more than 100 m away from the disturbance track. Thinner blanketing < 0.26mm was observed over an area of ~3km length and ~2.5km width around the disturbance (Yamazaki and Kajitani 1999), covering a much wider area compared to the *Kriging* approach.

In 1995, the InterOceanMetal (IOM) Joint Organization conducted a benthic disturbance experiment (IOM-BIE) over an area of 2000 x 1500m also in the eastern CCFZ, once more using the DSSRS (Kotlinski and Stoyanova, 1999; Radziejewska, 2002). Studies focused on the physical and chemical properties of the re-suspended and resettled sediments rather than on the spatial distribution of the material; this leads to only limited information on the amount of re-suspended material. Radziejewska (2002) estimated the volume of re-suspended material to be approximately 1800 $m^3$ over the entire duration of the experiment, but the actual volume is not known.

In 1997, the "Indian Deep sea Environment Experiment" (INDEX) was carried out in the Central Indian Ocean Basin. For the fourth time, the DSSRS was used to create the disturbance during 9 days of operation (Desa, 1997; Sharma and Nath, 1997). Results from sediment traps distributed up to 800 m away from the track show an increase in average particle fluxes from 48 to 150 mg $m^{-2}$ $d^{-1}$ during the disturbance phase. The flux decreased to 95 mg $m^{-2}$ $d^{-1}$ within the first six days after the disturbance stopped (Sharma, 2001). Based on visual observations, most of the sediment particles re-settled already within 150 m from the edge of the disturbance area (Sharma et al., 2001), with the major part of material settling within approximately 100 m distance (Sharma, 2000).

The last large-scale BIE was conducted in 1997 by MMAJ within the area of the Marcus-Wake Seamounts in the North Pacific Ocean (Yamada and Yamazaki, 1998). The induced sediment plume was visually monitored (Yamazaki et al., 1999) and data revealed a sediment blanketing thickness on top of Mn-nodules of up to 0.2 mm (Yamazaki et al., 2001). Due to the different geological setting (seamount in 2200 m water depth) and different sediment properties (calcareous sediments, coarser sediment particles, stronger currents), these results are not directly comparable to the results from most of the other BIEs mentioned above.

Reviewing the different large-scale BIEs and pilot mining tests conducted between the late 70's and late 90's it becomes obvious that the different experimental setups and the missing uniform definition of 'a' plume (grain size distribution, flocculation behavior, total mass per liter, settling velocity etc.) make it impossible to use the presented information for a meaningful predict of the behavior of a sediment plume created during a real deep sea mining operation (Peukert et al., 2018). As basis for sample interpretation thus reconstructing the initial disturbance of 1989 in the DISCOL area, which is considered as the most extensively sampled and monitored BIE site, might help to gain new and more conclusive insights in terms of the distribution of re-suspended and re-deposited sediment during and shortly after conducting the disturbance.

This study presents new data from the DEA, which were acquired in 2015 during RV SONNE cruise SO242-1 with state-of-the-art AUV multibeam and side scan sonar systems, cameras and under water navigation technology (Greinert, 2015).

## 1.3 DISCOL revisited in 2015 and objectives of this study

Since 1989, major technological advancements improved deep sea investigations with regard to data acquisition technologies and positioning accuracy. In 1989 GPS for example was not as sophisticated and high-resolution acoustic seafloor mapping with multibeam echosounder systems (MBES) was not as developed as it is today (e.g. 59 beams compared to 432 beams; single swath compared to dual swath; Lurton, 2017). AUV-based technologies did not even exist.

To acquire most accurate data of the old plough tracks, the entire DISCOL area was re-mapped using ship- and AUV-based hydroacoustic MBESs with different resolution (Boetius, 2015; Greinert, 2015). This provided new information for reconstructing the extent and impact of the initial disturbance experiment, the different geological settings within and next to the DEA and related varying and habitats. The results presented in this study mainly focus on the data collected by GEOMAR's AUV ABYSS (Linke and Lakschewitz, 2016, http://dx.doi.org/10.17815/jlsrf-2-149). The AUV was deployed in three different modes running either MBES, side scan sonar (SSS) or a photo camera system enabling autonomous mapping with a resolution of 2 m for bathymetric data, 0.5 m for SSS data and a few mm per pixel for photo surveys. All systems show clear evidence of the disturbance tracks created by the plough-harrow 26 years before.

This study presents the best georeferenced data set of the study area through a combined processing of the available ship- and AUV-obtained acoustic and optical data. In addition to this mapping exercise the succession of the disturbance tracks as well as their correct location is reconstructed, as this could not accurately be documented in 1989. Although the 78 plough tracks were created over a period of only 4 weeks (Thiel and Schriever, 1989) a more detailed understanding of their sequence is relevant regarding faunal differences from within or close to plough tracks in strong or weaker disturbed parts of the DEA. Furthermore for the understanding of varying down-core geochemical gradients the spatial thickness-change of the resettled sediment, the "blanketing", needs to be understood. This thickness distinctly differs between the plough tracks depending on if they were created in an earlier or later stage of the disturbance, which highlights the difference between high plume-sedimentation rates and natural deep-sea low sedimentation rates. Next to this an unbiased and correct comparison between areas that have not been impacted by any re-settled sediment with areas that have been impacted to various amounts, should be performed. Interpreting biological or geochemical results correctly requires a very precise knowledge of the exact and absolute sample or footage location on the seafloor and their spatial relation to the tracks which are only a few meters wide and apart from each other. Thus a correct georeferencing of all different data layers was a significant task of this study and although highly developed positioning systems were used in 2015 uncertainties and deviations of tens to a few hundreds of meters occurred. This task became even more important for georeferencing legacy data from 1989 for conclusively defining changes between 1989 and 2015 and spatial sediment re-settling differences established already during the plough experiment.

## 2 Data and Methods

### 2.1 Digitizing and archiving of DISCOL legacy data

Until 2015, the location and path of the disturbance tracks as well as the position of video and photo material of the past OFOS (Ocean Floor Observation System) surveys only existed as a vast collection of analogue (i.e. cruise reports, printed large navigational charts, video cassettes and slide films) and some digital records (i.e. OFOS annotation files, sample analysis as text or EXCEL files, e.g. Bluhm, 1994, Bluhm and Thiel 1996, Thiel and Schriever 1989, Schriever 1990, Schriever and Thiel 1992, Schriever et al., 1996). In preparation for the 2015 cruise, these records were digitized and compiled in a database, also including all other available sampling stations (i.e. BC, MUC, moorings, baited traps see e.g. Drazen et al., 2019) that were done during the first four expeditions to the DEA. This database was used for station planning prior to the SO242 cruises and allows comparing past and present disturbance levels and seafloor and ecosystem conditions at their best possible correct location.

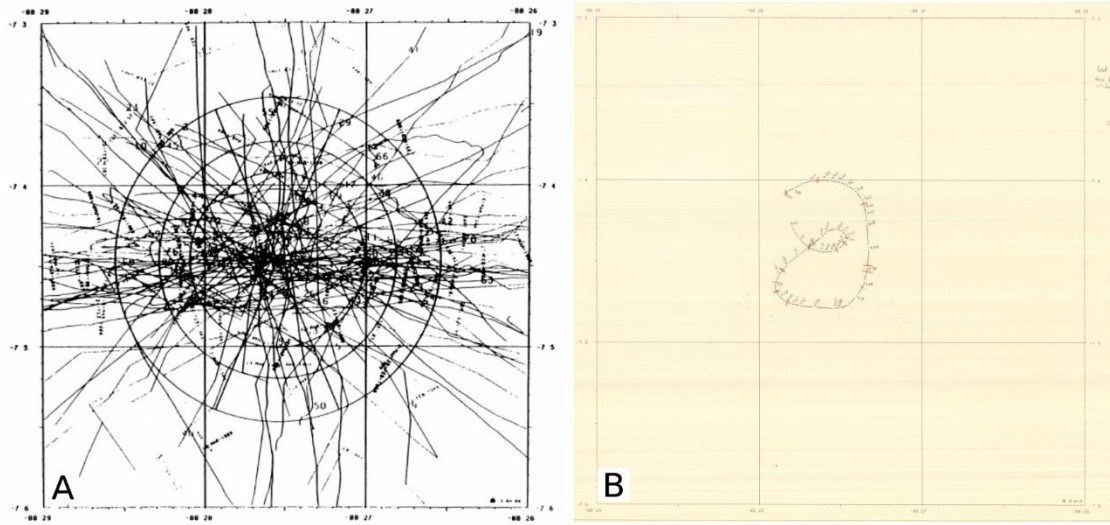

**Figure 1: Legacy data from the first cruise (SO61) to the DEA: A) Reported location of the disturbance tracks in 1989 (modified from Thiel and Schriever, 1989); B) Print of the navigation records of OFOS009 during SO061.**

### 2.2 Hydroacoustic and optical data acquired during cruise SO242_1

#### 2.2.1 Data acquisition

Acoustic and optical data were collected in 2015 during cruise SO242-1 with the German R/V SONNE (Greinert, 2015). Large-scale bathymetric data were acquired by the hull-mounted Kongsberg EM 122 MBES (12 kHz, 1° by 0.5° beam angle, 432 beams, equidistant, processed with QPS Fledermaus) already on board the vessel. The system was run with a swath angle of 130° at a survey speed of about 8knots (kn). The deployed AUV (for MBES, SSS and photo surveys) is a REMUS 6000 type AUV (Linke and Lakschewitz, 2016) equipped with a RESON Seabat 7125 MBES (200 kHz, 1° by 2° beam angle) and an Edgetech 2200 MP side scan sonar system (120kHz). The MBES surveys were conducted at an altitude of 80 m (Abyss192 - SO242/1_047-1; Abyss193 - SO242/1_060-1; Abyss194 - SO242/1_069-1; Abyss195 - SO242/1_075-1). During

the SSS surveys the altitude of the AUV was set to 40 m (Abyss188 / SO242-1_18-1) and 20 m (Abyss189 - SO242/1_25-1; Abyss190 - SO242/1_33-1). The AUV camera system "DeepSurveyCam" (Kwasnitschka et al., 2016) was used during 10 photo surveys (Greinert, 2015; Simon-Lledó et al., 2019) between 4 m and 9 m altitude and a at mean speed of 3 kn. More than 50,000 usable images were recorded (Greinert et al., 2015) and analyzed in terms of nodule coverage and size by automated image analysis (Schoening et al., 2017). Two photo

mosaics have been created from the AUV camera surveys Abyss196_SO242/1_83_1 (photos acquired in 7 m altitude) and Abyss199_SO242/1_102_1 (photos acquired in 4.5 m altitude).

Additional visual investigations during all cruises to the DEA were conducted using the towed camera system OFOS either equipped with both a still and video camera (Bluhm and Thiel, 1996, Thiel and Schriever, 1989; Schriever, 1990; Schriever and Thiel, 1992; Schriever et al., 1996) or just a video camera, which was mounted

on the frame of a sampling device (Boetius, 2015; Greinert, 2015). The DEA was crossed by a total of 55 successful OFOS surveys, during SO61 (16 surveys – navigation data for OFOS002 and OFOS012 are missing; Thiel and Schriever, 1989), SO64 (7 surveys; Schriever, 1990), SO77 (7 surveys; Schriever and Thiel, 1992), SO106 (7 surveys; Schriever et al, 1996), and SO242 (18 surveys; Boetius, 2015; Greinert, 2015). Ship-based Ultra-Short Baseline underwater navigation (USBL) was used for OFOS deployments during almost all cruises

with lower accuracy during three of the initial cruises (SO61, 77, 106); it failed during SO64 (Schriever, 1990).

### 2.2.2 Data description / working area

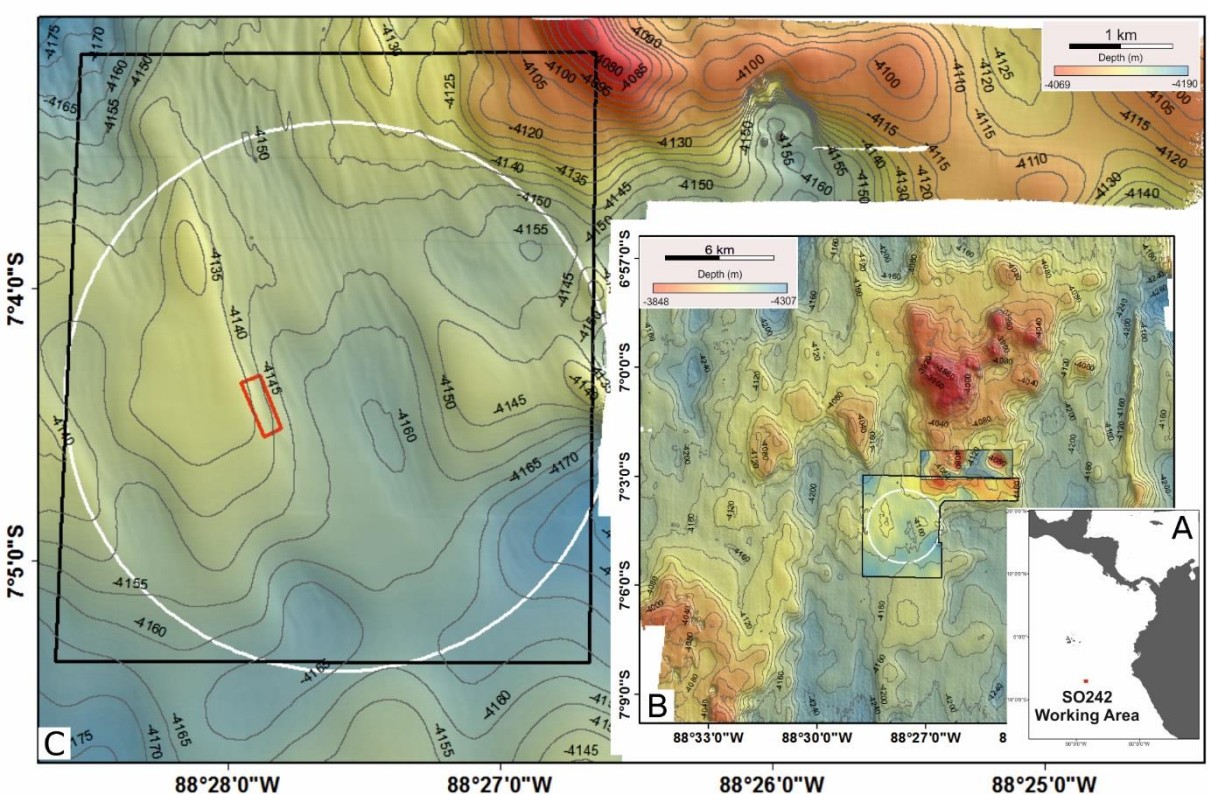

**Figure 2: (A) Location overview of the DISCOL area (red square; coastline shape-file from Wessel and Smith, 1996). (B) Ship-acquired bathymetric map of the working area from SO242 (white**

**circle marks the DEA; black polygons mark the area mapped by the AUV). (C) AUV-acquired bathymetric map covering the DEA and a part of the hilly area NE of the DEA (merged from**

**Abyss192 - SO242/1_047-1; Abyss193 - SO242/1_060-1; Abyss194 - SO242/1_069-1); black polygon indicates the SSS mapped area; white circle marks the DEA; red rectangle marks the location of the AUV acquired photo mosaic (Figure B2).**

The working area of cruise SO242 includes the DEA in the center and extends about 10 – 13 km around it. Generally the area is located about 800 km West of the Peruvian coast and about 700 km South of the Galapagos Islands (Figure 2A). N-S striking graben and horst structures can be seen throughout the entire area, corresponding to the highest slope angles of up to 36° (Fig. B3B); they are related to the tectonic setting of the study site being located on the Nazca Plate which originates from the East Pacific Rise (Devey et al., in review).

Within the working area the water depth varies between 4300 m and 3850 m (Figure 2B), with the minimum water depth corresponding to the summit of a rough sloping (>30° slope angle, Fig. B3B) seamount (rising ~200 m) north of the DEA (Devey et al., in review). West of the summit the terrain drops along one of the NS-striking graben structures with two lower sea mounts of about 100 m height. About 18 km to the SE of the DEA another larger seamount rises up to 3980 m water depth showing pit structures of tens of meters in depth and width as

has been recently described for the wider region to be generally associated with hill crests (Devey et al., in review). In the very west of the working area a NS-striking narrow ridge highlights again a tectonic nature of the area with another element of the graben and horst fault system in the area. Besides of these dominating bathymetric features, the rest of the terrain shows smooth undulating elevations and basins of several tens of meters depth and few kilometers width, with slope angles of <10°.

The finer structure of these flatter parts is much better resolved in AUV-acquired MBES data (Abyss192-194). The gently sloping terrain exhibits up to 15 m high hill/ridge and basin structures in the DEA and the western part of the mapped area (Figure 2C, Figure B4A). In 99 % of the DEA the maximum slope of the terrain is only 3° (Figure B4B) with generally NNW-SSE striking morphological features. Parallel to these, < 1 m high and 20-40 m wide ripple structures extending from the center of the DEA towards the North. The appearance of these

features remind of ripple structures oriented parallel to the predominant bottom current direction in this area (Thiel and Schriever, 1989, Greinert, 2015) and are further described below within this section. In the NE the terrain rises distinctly forming up to 50 m high summits (Figure 2C, Figure B4A, Fig. B3). Within this mountain area an approximately 50 m deep circular crater structure can be seen (Figure 2C) that is surrounded by steep slopes of up to ~50° (Figure B4B). Within the crater two ~10 m high conical hills consist of pillow basalts

(revealed by OFOS footage; SO242-1_#135_OFOS6), as a result of sub-recent volcanism in the area (see Devey et al., in review).

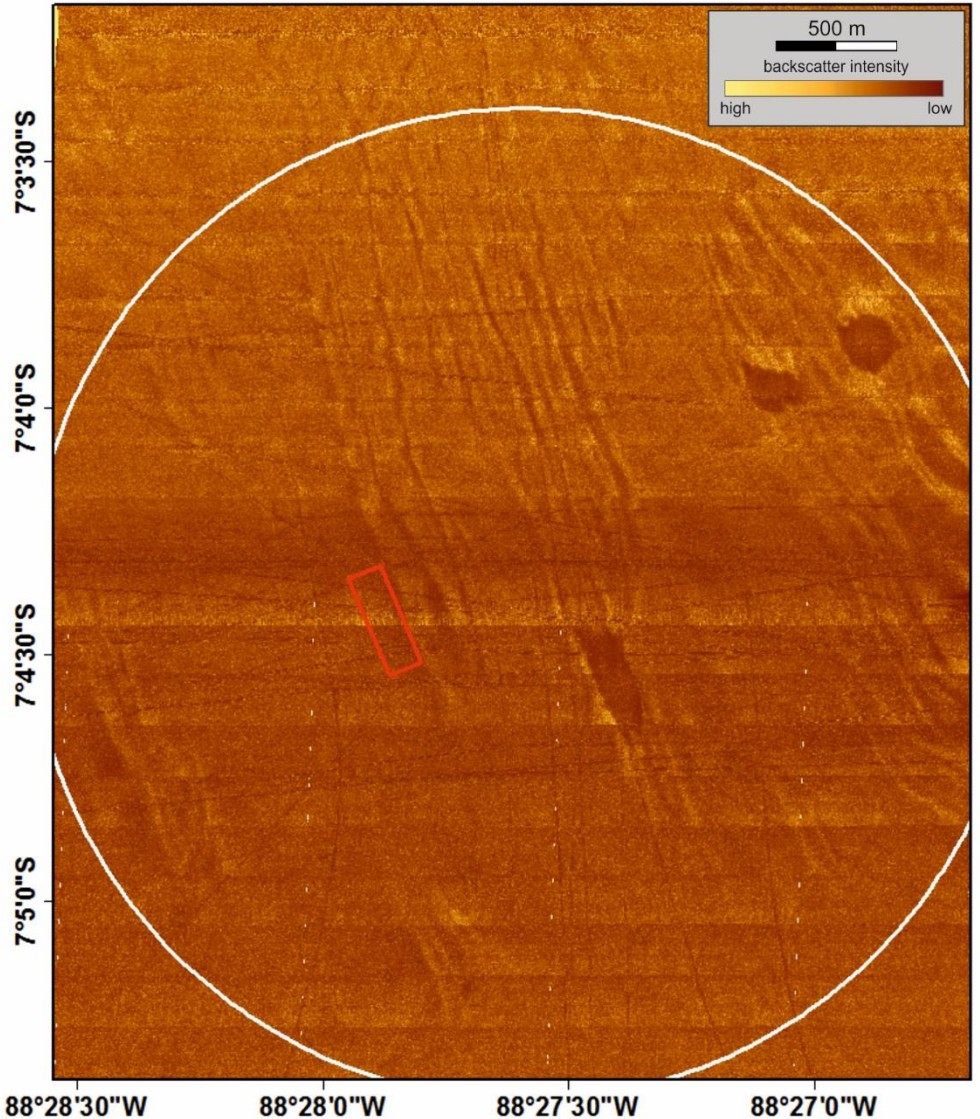

**Figure 3: AUV-acquired SSS map (resolution 50cm) including the DEA (white circle); bright colors indicate high backscatter signals, dark colors indicate low backscatter signals; red rectangle indicates the location of the photo mosaic (Figure B2).**

Using the AUV side scan sonar (SSS) an area of 4 x 3.5 km with the DEA in the center was mapped (Figure 3). The acoustic signals captured a significant amount of the plough tracks, which appear darker in the SSS map, representing a lower backscatter. Three dark distinct patches between 140 and 200 m in size are apparent within the side scan data, indicating softer substrate within these structures that bathymetrically represent sediment filled local basins of ~5 m depth with a rather horizontal seafloor. The MBES backscatter and side scan sonar data of the NNW-SSE- striking channel structures, indicating deposition of softer sediment within the depressions. The channel structures are oriented parallel to the prevailing strong bottom current direction within the area towards the NNW (Thiel and Schriever, 1989; Schriever and Thiel, 1992) and the undulating shape of the structures indicate a generation by a flow regime and not by tectonic activities, which would appear straighter. We assume that bottom currents are channelized through the local trough around the rising terrain towards the NE and may cause turbulent flows which eventually cause furrowing. This process has been

described also in the deep ocean with a dominant strong bottom current flow between 5-20 cm –s (Flood, 1983), which is given in the DEA and hence this process could have formed these structures.

**2.3 Geo-referencing of AUV data sets**

AUV operation in great water depth suffers from inaccurate positioning of acquired data sets. Underwater positioning is typically determined using hydro acoustic techniques as ultra-short base-line (USBL, measurement between the ship and the AUV) or long-base-line systems (LBL, triangulation of the AUV using seafloor deployed transponders). AUV "Abyss" navigates autonomously using a combination of different navigational methods (Linke and Lakschewitz, 2016). During our studies LBL navigation was only used to set an accurate starting position of the AUV at the beginning of each survey after arriving at the seafloor. No additional LBL fixes were considered as this often results in abrupt track corrections that cause unwanted artifacts, particularly in SSS data. Instead, navigation after the initial LBL fix relied on a Doppler Velocity Log (DVL) data, inertial navigation sensors and dead reckoning data fusion as supplied by the AUV system (Linke and Lakschewitz, 2016). Typically, such kind of navigation is prone to slow drifts, which over the course of an entire mission (up to 20 hours operation time) can add up to several tens or hundreds of meters offsets. These navigational shifts need to be derived and corrected during processing when comparing or combining several different data sets as MBES, SSS, and imagery of the AUV, imagery of OFOS and ship-based bathymetry

To achieve the best possible alignment and absolute geo-referencing, the ship-based EM122 bathymetric data with a spatial resolution of 38 m were taken as absolute reference layer (Fig. C1). The AUV bathymetric data with a spatial resolution of 2 m was resampled to match the 38 m resolution enabling a direct grid comparison (e.g. grid subtraction, Fig. E1) and correction of vertical and lateral offsets of the AUV bathymetric grid relative to the ships data layer. Using 5 m contour lines to visualize morphological features in the area, the 38 m AUV bathymetry was shifted/stretched manually onto the EM122 data (Figure 4). Subsequently, the high-resolution AUV bathymetric digital terrain model was shifted in the same way using the ArcGIS 10.2 *Georeferencing Toolbox* for geographic corrections (contour lines were derived with the *Spatial Analyst Toolbox* and grids were subtracted to see z-offsets using the *Raster Calculator* function). Fig. 4 shows the high-resolution AUV bathymetry with the contour lines derived from the ship-based bathymetric grid to visualize the accordance of both data sets after the alignment.

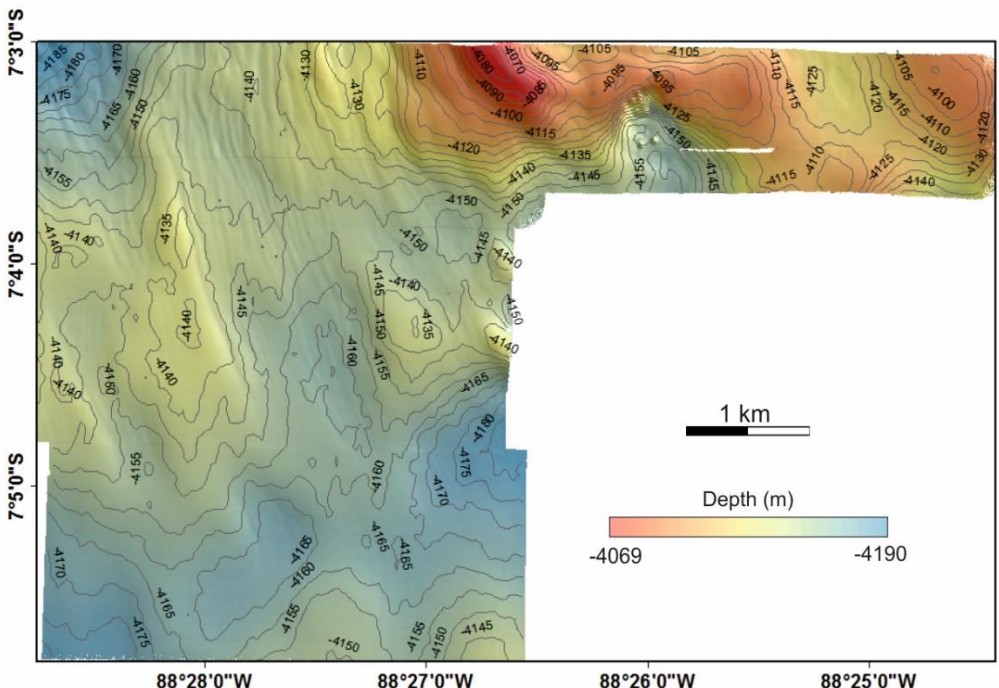

**Figure 4: AUV-based bathymetry after alignment to the ship-based bathymetry. The plotted contour lines are the 5 m contours of the ship-obtained MB data set.**

The SSS map was georeferenced relative to the AUV MBES data using a number of disturbance tracks, visible in the bathymetry (Figure 5A, B and C) and the SSS data (Figure 5D), and three prominent Mn-nodule-free depressions, which appear distinctly dark in the SSS map (Figure 5C), as anchor features.

Based on the same structures the photo mosaics of the DEA could be aligned to the SSS map (Figure 5E). Finally, visually detectable sampling locations of BC, GC, or MUC impacts were used to validate the accuracy of the geo-referencing by comparison with their actual USBL positions (see Appendix D for details).


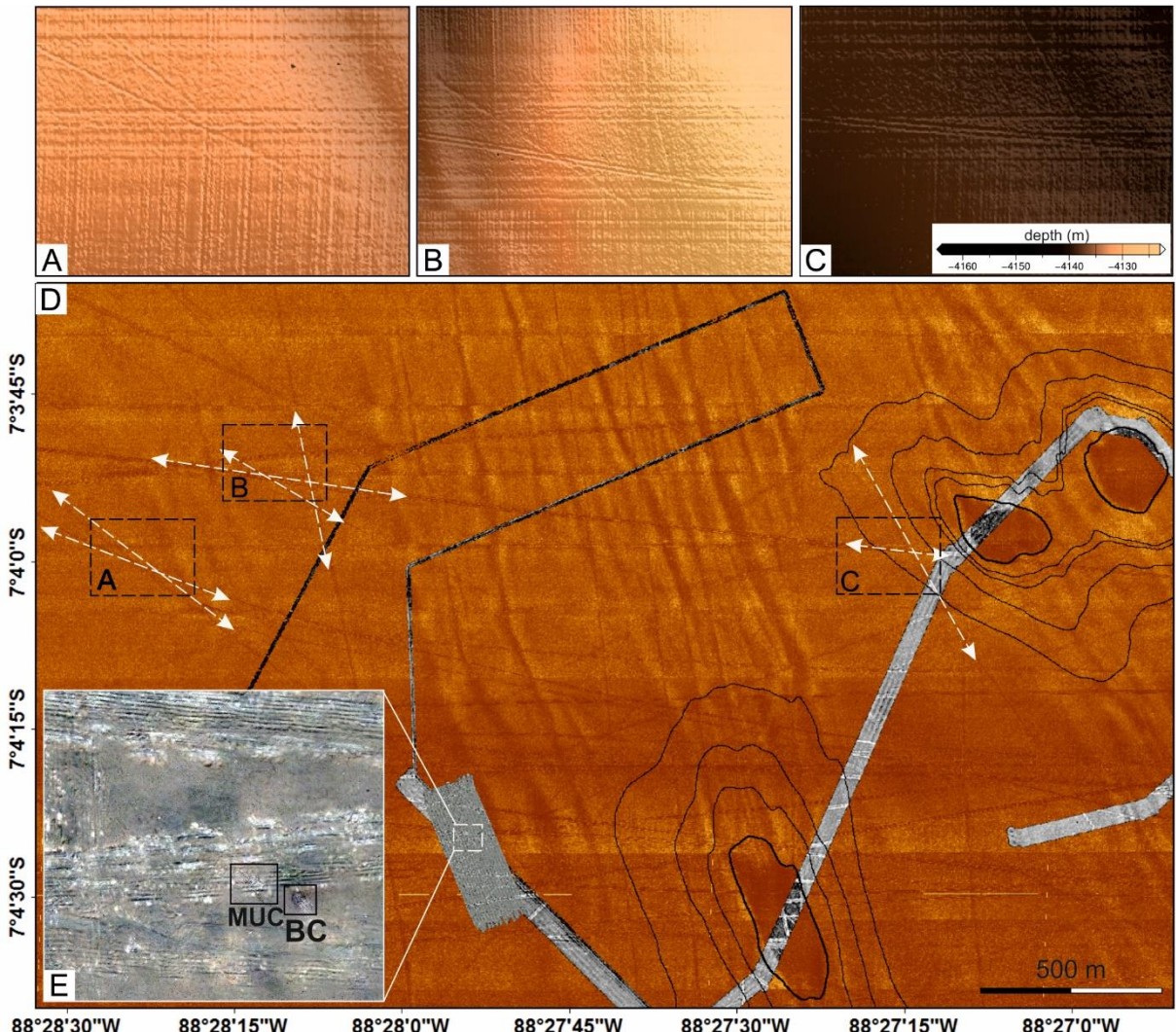

**Figure 5: Plough marks visible in the high resolved bathymetric (A, B, C) and the SSS map (D), as well as three characteristically shaped depressions well visible in both data sets (D, black lines mark local 1 m contours) were used for the alignment of AUV MB and SSS data (see Fig. C2 for a larger section of the AUV MB map). The mosaics from AUV-acquired seafloor images (grey-colored in D) were linked to the SSS map based on the same structures. The elongated photo survey is colored by Mn-nodules m$^{-2}$ (Schoening et al. 2017) within the photos where the nodule-free areas appear distinctly dark; this was used for the alignment of the different data sets along the three "dark patches". Sampling locations visible in the photographs (E) function as anchor points to evaluate the referencing accuracy by the comparison with their USBL position.**

### 2.4 Position and age sequence of disturbance tracks

Disturbance tracks visible in SSS data (Figure 5D) were manually digitized using functionalities of ArcGIS. Each track was given a unique identifier and was assigned to one of four classes reflecting the general orientation of the respective track: *H* for E-W orientated tracks, *D* for NW-SE and NE-SW orientated tracks, *V* for N-S orientated tracks and *P* for non-continuous tracks and track segments (H=horizontal, D=diagonal, V=vertical, P=parts of tracks; Figure 6; Table 1). The track IDs were arbitrary given during the digitizing and were not renamed after the sequencing, thus the numbers do not reflect the age sequencing. During the digitizing

it occurred that some of the tracks labeled as P=parts are as long as others assigned to one of the "entire track-groups (V, H, D)", due to subsequent extension after further investigations. Because of this there is no clear definition when tracks are labeled P; however, typically "P" tracks are shorter than 1200 m.

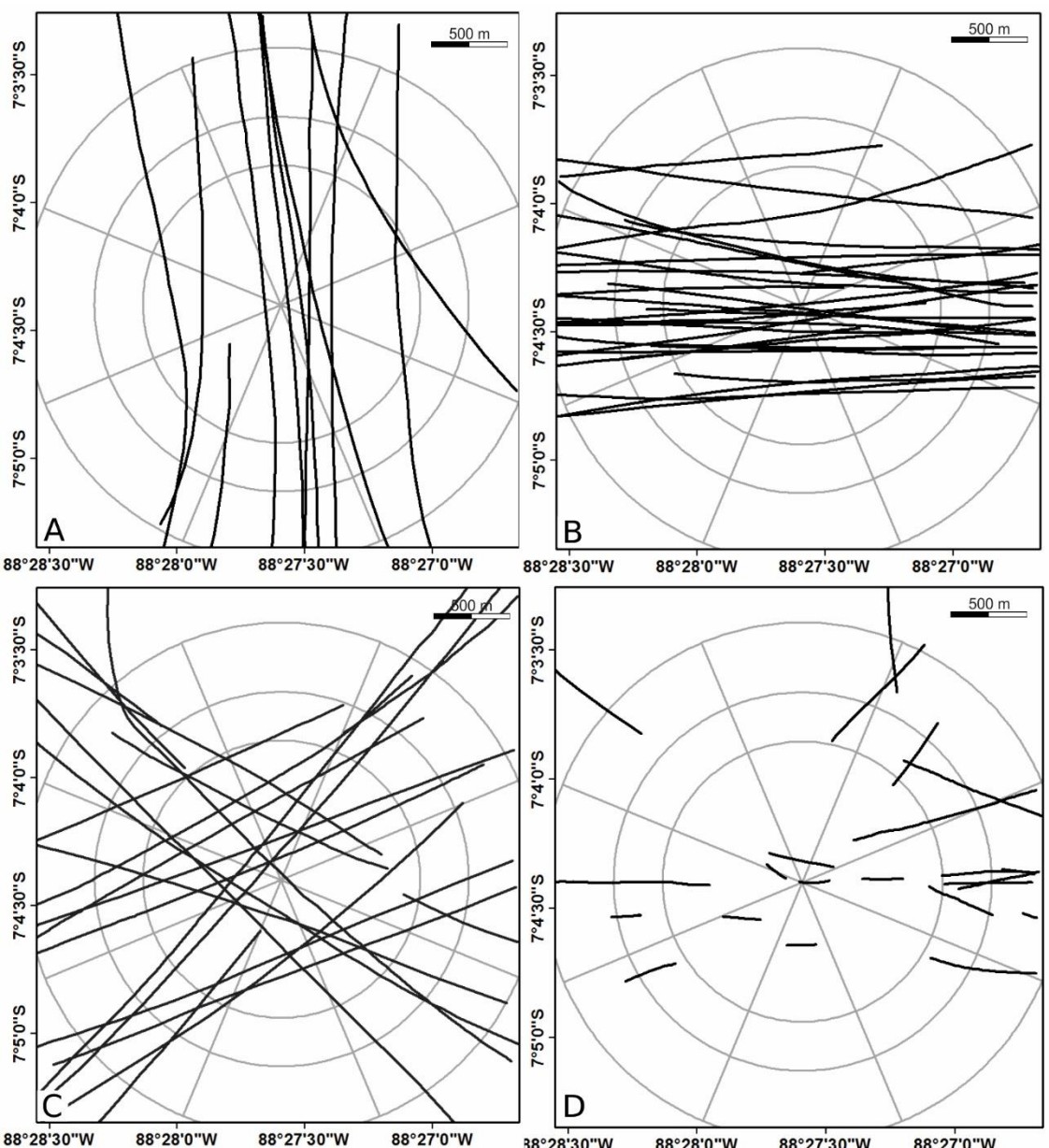


**Figure 6: Identified plough tracks classified and named after their orientation within the DEA (grey circled area); A) 11 vertical tracks "V"; B) 28 horizontal tracks "H"; C) 21 diagonal tracks "D"; D) 24 partial tracks/segments "P".**

Generally during the ploughing in 1989 several tracks were undertaken during one deployment of the plough

(station name PFEG-1 to PFEG-11; PFEG1 was a gear handling test a few nautical miles south of the DEA). After the first two groups PFEG2 and PFEG3 OFOS dive OFOS009 and OFOS010 were conducted during SO61 and the photo and video material collected during these two OFOS dives could be examined for track occurrences. The track orientation was determined from each seafloor image and matched to the track orientation on the SSS map considering the course over ground (COG) and heading of the OFOS (Figure 7) to distinguish

the correct plough track. This way and considering the log files from cruise SO61, which gave an idea about rough course and location of the ploughs, the tracks corresponding to the first two plough groups (15 tracks) could be identified. Considering photo and video material from ROV (http://dx.doi.org/10.17815/jlsrf-3-160), AUV and OFOS surveys and the SSS data, intersections were visually examined to establish a relative age succession between the investigated tracks (Figure 8).

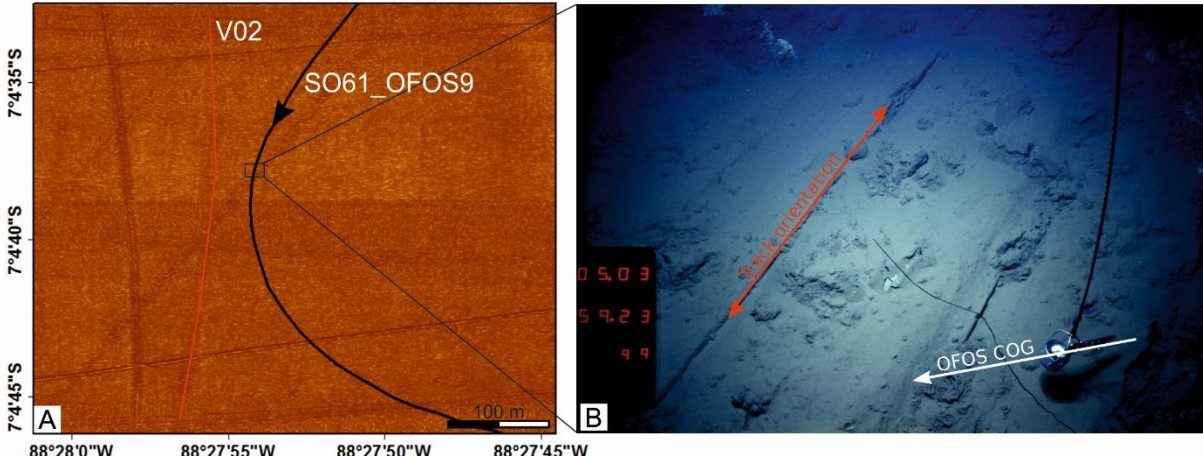

**Figure 7: Identifying the very first disturbance tracks (PFEG2) using the reconstructed instrument navigation (SO061_OFOS9) plotted on the SSS map of the DEA (A) and camera data collected during this survey (B) with respect to the track orientation and the previously determined relative age succession.**

For some intersections the sequence could not directly be established and was inferred considering the relative track age information of other intersecting tracks (Figure 8). Unfortunately, this workflow could not be applied for the later groups of tracks (PFEG4 to PFEG11) since they were not directly followed by an OFOS survey. Thus the track density until the next OFOS observation got too high and considering the navigation uncertainties, an unambiguous assignment of the tracks is not possible. The reconstruction of the age succession

of all tracks was finally done using a 84x84 matrix (including 60 identified tracks and 24 track segments, Table F1) where all observed crossings were included. Logical process of elimination and cross-referencing of individual tracks relative to all other tracks in combination with their position and the reconstructed ships navigation during the time of the experiment (Fig. 1A) was performed. Based on this, the tracks were assigned to their respective PFEG and to track ID's (Table 1).

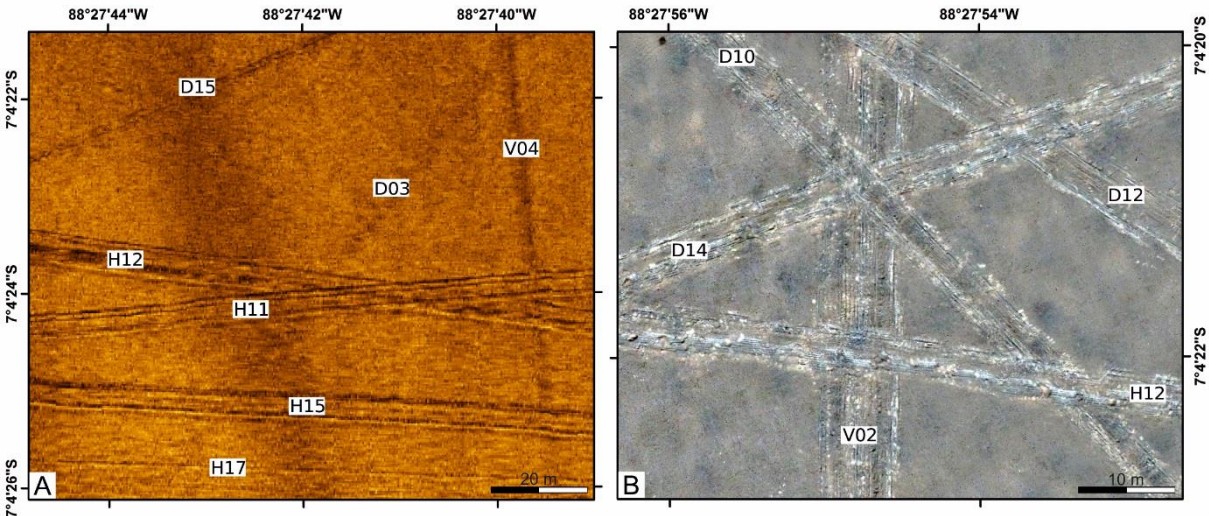

**Figure 8: Establishing the relative age sequence based on intersections between two tracks from the SSS map (A) and seafloor photographs (B). Absolute age information can be derived from cross-referencing relative age information of more than two individual tracks (B). Here the age sequence of all shown tracks is V02 < D12 < D14 < D10 < H12.**

*2.5 Reconstructing the impact of the re-settled plume*

The initial impact of the plough tracks is given through the mixing (ploughing) of the top 20 to 30 cm of the sediment and the related suspension of sediment into the bottom water (Foell et al., 1990). Nodules were not removed from the seafloor but ploughed under (Thiel and Schriever, 1989). The re-sedimentation of the initiated sediment plume is considered the secondary impact. For reconstructing the initial impact and the proximal (in images visible) sediment blanketing the course of the plough tracks were used in combination with bottom current information recorded during the time of the experiment to establish a disturbance intensity map (including initial and secondary impact). Considering also the plume deposition information from other BIEs (Table A1) and the recent study by Peukert et al. (2018), the qualitative sediment blanketing thickness within the DEA was determined based on the following assumptions and set parameters. Each track was assumed to have a width of 8 m, not considering the possible handling problems with the plough-harrow (e.g. being towed only on the side, short loss of bottom contact, Thiel and Schriever, 1989). The intensity of the disturbance was assumed to be the highest within and close to the tracks and the sediment blanketing thickness to decrease with increasing distance off the track. Studies from other BIEs showed visual sediment blanketing distances between 70 m and 150 m in current direction away from the track. It is assumed that the majority of the re-suspended sediment (about 90 %) resettled over this distance (Lavelle et al., 1981; Peukert et al., 2018).

The main factor controlling the re-deposition are current speed/direction and particle settling velocity with the latter being describable as a function of the particle size according to Stoke's Law and the method described by McCave (McCave, 1984; Jankowski et al., 1996). The sediments within the DEA are composed of layered clayey silts or silty clays, with a sand fraction of ca. 5 % consisting of foraminiferous residues and shell fragments (Grupe et al., 2001). According to Lavelle et al. (1981), Schriever et al. (1996) and Becker et al. (2001) the stirred up sediment mixture induced flocculation and aggregation of particles causing a very rapid re-sedimentation ($\geq 1$ cm s$^{-1}$) of the plume within the first 20 m away from the track. Latest research on mining-induced sediment plume-behavior also indicated a near-track heavy sediment blanketing (Gillard et al., 2019).

The sediment blanketing decreases as a function of reduced particle settling velocities as finer particles dominate the plume composition and stay longer in suspension (Lavelle et al., 1981).

Bottom current direction and velocity determine the direction of the re-sedimentation area and sediment spreading (Lavelle et al., 1981; Jankowski et al., 1996; Greinert, 2015). Bottom currents in the Central Pacific are reported to be distinctly different even at locations only a few kilometers apart (Robinson and Kupferman, 1985). Several measurements in the DISCOL area revealed a predominantly northern to northwestern direction with maximum current speeds of 17 cm s$^{-1}$ (Thiel and Schriever, 1989; Schriever and Thiel, 1992) indicating a transport of the re-suspended particles primarily in this direction. The undertaken measurements showed that the currents in the DEA alternate between strong (> 5 cm s$^{-1}$) and quasi unidirectional currents towards the NNW and weaker currents (< 1-3 cm s$^{-1}$) with greater directional variability (Klein, 1993; Klein, 1996). This variability has also been observed during the first cruise SO61 to the DISCOL area (Thiel and Schriever, 1989), with the "strong" current regime occurring during the first leg (February 1989) and the creation of PFEG1 to 7 and the weaker currents towards the end of the second leg (March 1989) and the creation of PFEG8 to 11, where the currents showed semidiurnal change of current direction from predominantly NNE to predominantly SSE. This certainly affected the sediment plume dispersal.

Since no information about the amount of re-suspended material is available, the impact is reconstructed qualitatively using values resembling disturbance intensity between 1 within the disturbance tracks and 0.1 representing the deposition of 90% of the re-suspended material at the maximum distance of the proximal disturbance. With regards to other impact monitoring results from large-scale disturbances (e.g. Lavelle et al., 1981, Table A1) and the results of small-scale disturbance experiments conducted during SO239 (Martinez Arbizu and Haeckel, 2015; Peukert et al., 2018), SO242/1 (Greinert, 2015), and SO242/2 (Boetius, 2015), the maximum distance affected by sediment blanketing was assumed to be 120 m *with*, and 20 m *against* the current direction for the "strong" current regime. These distribution values and a distribution direction of 334° was set for PFEG 2 to 7 and all recognized parts of plough tracks, which could not be assigned to a distinct PFEG. To account for the changing conditions during weaker bottom currents (PFEG 8 to 11), the distances were set to 100 m *with*, and 30 m *against* the current direction. Based on the statistics of the current directions (Thiel and Schriever, 1989) during the creation of 31 recognized tracks of that period, the plough tracks were divided in two groups, one considering a NNE-current (towards 18°, 19 tracks) and the other group considered a SSE-current (towards 143°; 12 tracks). Considering the semidiurnal current direction change, the assignment of the tracks to one of those groups was based on the estimation of the track creation time and duration in consideration of the length of one track and the speed of the ship while ploughing and the determined relative sequence of the single tracks.

For calculating the sediment plume deposition down-current and up-current (due to turbulences) the following simple function was used:

$$y = e^{-\left(\frac{x}{R}\right)}$$ [1],

with *y* representing the relative sediment thickness at distance *x* from the disturbance track.

An exponential function was chosen to account for the effects of flocculation and aggregation of the re-suspended sediment closer to the track. The factor *R* was introduced to meet the assumption that 10% of the re-

suspended material remains in the water column and being re-deposited at greater distances (Lavelle et al., 1981; Jankowski et al., 1996). This factor was considered for the particle transport *with* the prevailing bottom currents. *Against* the bottom currents the re-suspended material was assumed to completely resettled either within the first

20 m for strong currents or 30 m for weak currents. The relative sediment thickness was calculated in 0.8 m steps away from each disturbance track considering the above mentioned current directions. The final blanketing map was produced by adding all relative sediment thicknesses within each square meter of the DEA area using the *blockmean* command in GMT (argument –Ss to get the sum; Wessel et al., 2013) and producing an interpolated grid using the *nearneighbor* command. It is assumed that the plough intensity and sediment re-suspension did

not change during each plough track was created.

## 3 Results

### 3.1 Geo-referencing of data sets

Navigational offsets were detected between the different AUV missions with lateral offset between AUV- and ships data of 30 m to 80 m (Figure 4). As AUV datasets from four different MBES surveys are used, a good geo-

referencing of completely compiled AUV data set on the ships bathymetry was not possible. Therefore a focus for the best possible alignment was set to the DEA region with only three AUV data sets. To check for the improvement of the geo-referencing the AUV-bathymetry was subtracted from the ship-obtained dataset at identical resolution (Fig. E1). Prior to shifting and stretching of the AUV grid, the depth differences showed a mean offset of -9 m. Thus 9 m were added to the entire AUV bathymetric grid to account for this absolute z-

offset and after shifting/stretching, the difference between the AUV and ships bathymetry showed the mean to be at 0 m depth-difference with only +-0.5 m median range (Fig. E1).

As for the MBES data, the lateral offset between different SSS data surveys was not constant but varied between 40 m and 50 m. Geo-referencing the combined SSS map onto the AUV-bathymetry showed offsets between 30 and 80 m that were corrected (Fig. G1). The photo mosaics, which could be aligned to the SSS map very

accurately, show sampling locations that we compared to the USBL position during the time of sampling for validating the geo-referencing results (Figure 9; Table G1). The mean difference between the georeferenced photomosaic sampling locations and those from the USBL navigation is 14 m (Table G1), whereas BC positions on average differ 11 m and MUC positions 19 m. These values indicate the overall absolute accuracy of the navigation and geo-referencing that could be achieved. The accuracy decreases in the outer regions of the SSS

data, as no additional information such as seafloor sampling locations or characteristic features are available.

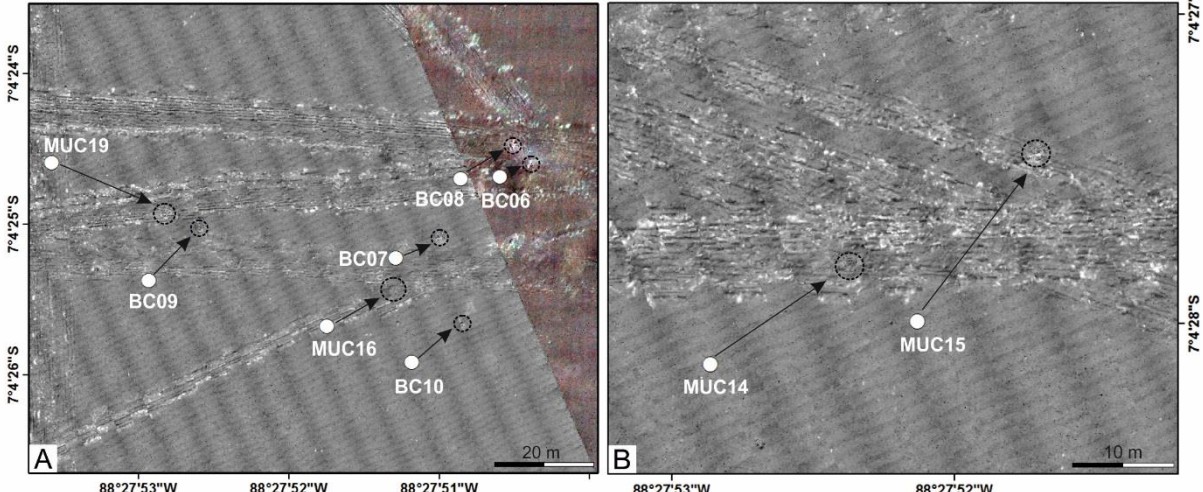

**Figure 9: Determining the accuracy of the geo-referencing based on impact features of seafloor sampling during SO242/1; black arrows indicate the offset between the sampling positions determined using USBL (white dots) and imprint on the seafloor (black circled) after applying the geo-referencing method.**

### 3.2 Plough tracks and their sequence

Within the SSS map a total of 60 continuous tracks were identified and assigned to three different classes: V (11), H (28) and D (21) (Figure 6). In addition, 24 track segments were found and represent the fourth class P (Figure 6; Table 1). Some of these partial tracks were assigned to other track identifiers (Table 1), based on the same course, but this could not be accomplished for all of the segments.

**Table 1: Identified plough tracks including the number of intersections and the absolute age with respect to the disturbance phase (PFEG2-PFEG11). Track ID arbitrary numbers that do not indicate any sequence, numbers are given during the digitalization.**

| # | sequence number | track_ID | number of intersections | PFEG | length (in m) |
|---|---|---|---|---|---|
| 1 | 1 | **D01** | 53 | 2 | 4746.47* |
| 2 | 2 | **D09** | 24 | 2 | 2866.55* |
| 3 | 3 | **V02** | 43 | 2 | 3411.77 |
| 4 | 4 | **D07** | 33 | 2 | 3281.15* |
| 5 | 5 | **P01** | 1 | 2 | 1013.31* |
| 6 | 6 | **V03** | 14 | 2 | 1559.89* |
| 7 | 7-29 | **D20** | 31 | 3/4/5 | 3580.93* |
| 8 | 7-30 | **D05** | 6 | 3/4/5 | 1608.99* |
| 9 | 7-31 | **D08** | 22 | 3/4/5 | 2465.54* |
| 10 | 7-31 | **P02** | 5 | 3/4/5 | 957.88 |
| 11 | 7-32 | **H07** | 15 | 3/4/5 | 1673.07* |
| 12 | 7-32 | **H16** | 25 | 3/4/5 | 2738.12* |
| 13 | 7-32 | **H21** | 28 | 3/4/5 | 3538.1* |
| 14 | 7-32 | **P10** | 5 | 3/4/5 | 584.55* |
| 15 | 7-38 | **P17** | 7 | 3/4/5/6 | 1117.16 |
| 16 | 7-46 | **P27** | 3 | 3/4/5/6/7 | 170.68 |
| 17 | 7-46 | **P31** | 1 | 3/4/5/6/7 | 147.82 |
| 18 | 7-46 | **P30** | 7 | 3/4/5/6/7 | 359.03 |
| 19 | 7-46 | **P29** | 9 | 3/4/5/6/7 | 584.54 |

| 20 | 7-46 | **P19** | 4 | 3/4/5/6/7 | 267.84 |
| 21 | 7-46 | **P21** | 2 | 3/4/5/6/7 | 766.90* |
| 22 | 7-46 | **P22** | 1 | 3/4/5/6/7 | 379.54 |
| 23 | 7-46 | **P03** | 1 | 3/4/5/6/7 | 831.79* |
| 24 | 7-45 | **P05** | 7 | 3/4/5/6/7 | 1015.19* |
| 25 | 7-46 | **P07** | 7 | 3/4/5/6/7 | 472.93 |
| 26 | 7-46 | **D04** | 2 | 3/4/5/6/7 | 1597.6* |
| 27 | 8-30 | **D21/P04** | 18 | 3/4/5 | 1915.45*/552.3 |
| 28 | 8-46 | **P06** | 2 | 3/4/5/6/7 | 828.98* |
| 29 | 8-31 | **D06** | 3 | 3/4/5 | 1177.81* |
| 30 | 9-31 | **H26** | 22 | 3/4/5 | 2624.92* |
| 31 | 10-32 | **H02** | 18 | 3/4/5 | 2325.01* |
| 32 | 10-33 | **D18** | 39 | 3/4/5 | 3776.48* |
| 33 | 18-46 | **P15** | 4 | 4/5/6/7 | 626.25* |
| 34 | 18-46 | **P11** | 3 | 4/5/6/7 | 231.42* |
| 35 | 18-46 | **P14** | 3 | 4/5/6/7 | 500.08 |
| 36 | 20-39 | **D14** | 41 | 4/5/6 | 3721.02* |
| 37 | 20-39 | **D19** | 11 | 4/5/6 | 895.11* |
| 38 | 22-41 | **H22** | 29 | 4/5/6 | 2775.6* |
| 39 | 22-44 | **H25** | 25 | 4/5/6 | 3610.85* |
| 40 | 23-43 | **D12** | 47 | 4/5/6 | 4163.09* |
| 41 | 23-45 | **H24** | 18 | 4/5/6/7 | 2986.35* |
| 42 | 24-44 | **D13** | 39 | 4/5/6 | 3294.63* |
| 43 | 25-45 | **D16** | 39 | 4/5/6/7 | 3949.62* |
| 44 | 25-45 | **D10** | 50 | 4/5/6/7 | 4411.18* |
| 45 | 25-46 | **P23** | 3 | 4/5/6/7 | 212.86 |
| 46 | 26-46 | **D17** | 41 | 4/5/6/7 | 3626.51* |
| 47 | 47-48 | **H17** | 29 | 8 | 2782.15* |
| 48 | 47-48 | **H20** | 22 | 8 | 1717.86 |
| 49 | 49-50 | **D02** | 50 | 8 | 5248.51* |
| 50 | 49-50 | **D03** | 50 | 8 | 5099.99* |
| 51 | 51 | **D11** | 26 | 8 | 2224.16 |
| 52 | 52 | **V01** | 47 | 8 | 4290.25* |
| 53 | 53 | **H03** | 26 | 8 | 3545.04* |
| 54 | 54 | **H01** | 26 | 8 | 3486.25* |
| 55 | 55 | **H06** | 32 | 8 | 3524.81* |
| 56 | 56 | **D15** | 42 | 8 | 3598.8* |
| 57 | 57 | **V10** | 45 | 9 | 4219.36* |
| 58 | 58 | **V08** | 46 | 9 | 3800.9* |
| 59 | 59 | **V07** | 47 | 9 | 3943.95* |
| 60 | 60 | **V06** | 48 | 9 | 4299.11* |
| 61 | 61-63 | **V04** | 46 | 9 | 4247.57* |
| 62 | 61-63 | **V09** | 40 | 9 | 3432.81* |
| 63 | 61-63 | **V05** | 49 | 9 | 4238.12* |
| 64 | 64-65 | **P16** | 3 | 9 | 219.15 |
| 65 | 64-65 | **V11** | 41 | 9 | 3889.04* |
| 66 | 66-67 | **H18** | 33 | 10 | 3513.86* |
| 67 | 66-67 | **H11** | 32 | 10 | 3548.1* |
| 68 | 68-71 | **H10** | 26 | 10/11 | 3480.55* |
| 69 | 68-71 | **H13** | 31 | 10/11 | 3559.03* |
| 70 | 68-71 | **H09** | 27 | 10/11 | 3496.72* |
| 71 | 68-71 | **H23** | 32 | 10/11 | 3512.36* |
| 72 | 72-74 | **H19** | 28 | 11 | 3264.1* |
| 73 | 72-74 | **H04** | 29 | 11 | 3468.02 |
| 74 | 72-74 | **H08** | 30 | 11 | 3505.65* |

| 75 | 75-77 | **H05** | 25 | 11 | 3946.72* |
| 76 | 75-77 | **H12** | 34 | 11 | 3089.88* |
| 77 | 75-77 | **H28** | 25 | 11 | 3524.09* |
| 78 | 78-80 | **H14** | 18 | 11 | 2121.69* |
| 79 | 78-80 | **H15** | 31 | 11 | 3501.81* |
| 80 | 78-80 | **H27** | 24 | 11 | 3527.22* |
| 81 | - | **P13** | 0 | - | 289.68 |
| 82 | - | **P18** | 1 | - | 102.48* |
| 83 | - | **P20** | 0 | - | 227.23 |

\* tracks extend beyond the limits of the SSS data

## 3.3 Estimation of the impacted area

### 3.3.1 Initial plough impact

Based on the detected plough tracks (including the track segments) the directly impacted area is 1.9 km$^2$, corresponding to approximately 19 % of the DEA (10.81 km$^2$) assuming a width of each individual disturbance track of 8 m (Thiel and Schriever, 1989) and a length of approximately 3 km within the DEA. This area agrees with the original estimate (ca. 20%; Thiel and Schriever, 1989). However, this represents only an approximation of the disturbed area as the length of the tracks is variable and individual ones reach a length of up to 5 km and not all of the tracks could be identified to their complete extent. The disturbance tracks can clearly be observed to continue outside the target area of the 2 nmi in diameter DEA target-circle and the created impact on the ecosystem extends beyond the limits of the DEA and even beyond the area covered by the SSS data (Figure 10). The total plough area from 1989 is thus not exactly known.

In comparison the previously reported disturbance track locations and the observations from 2015 generally show the same trend with a high density of E-W oriented tracks and less tracks with N-S orientation (Figure 10). The locations of individual disturbance tracks do not agree well most likely because the plough tracks from 1989 were reconstructed from the ships position only (with a much lower accuracy than today) and an almost unknown layback of the plough-harrow behind the ship (Thiel and Schriever, 1989).

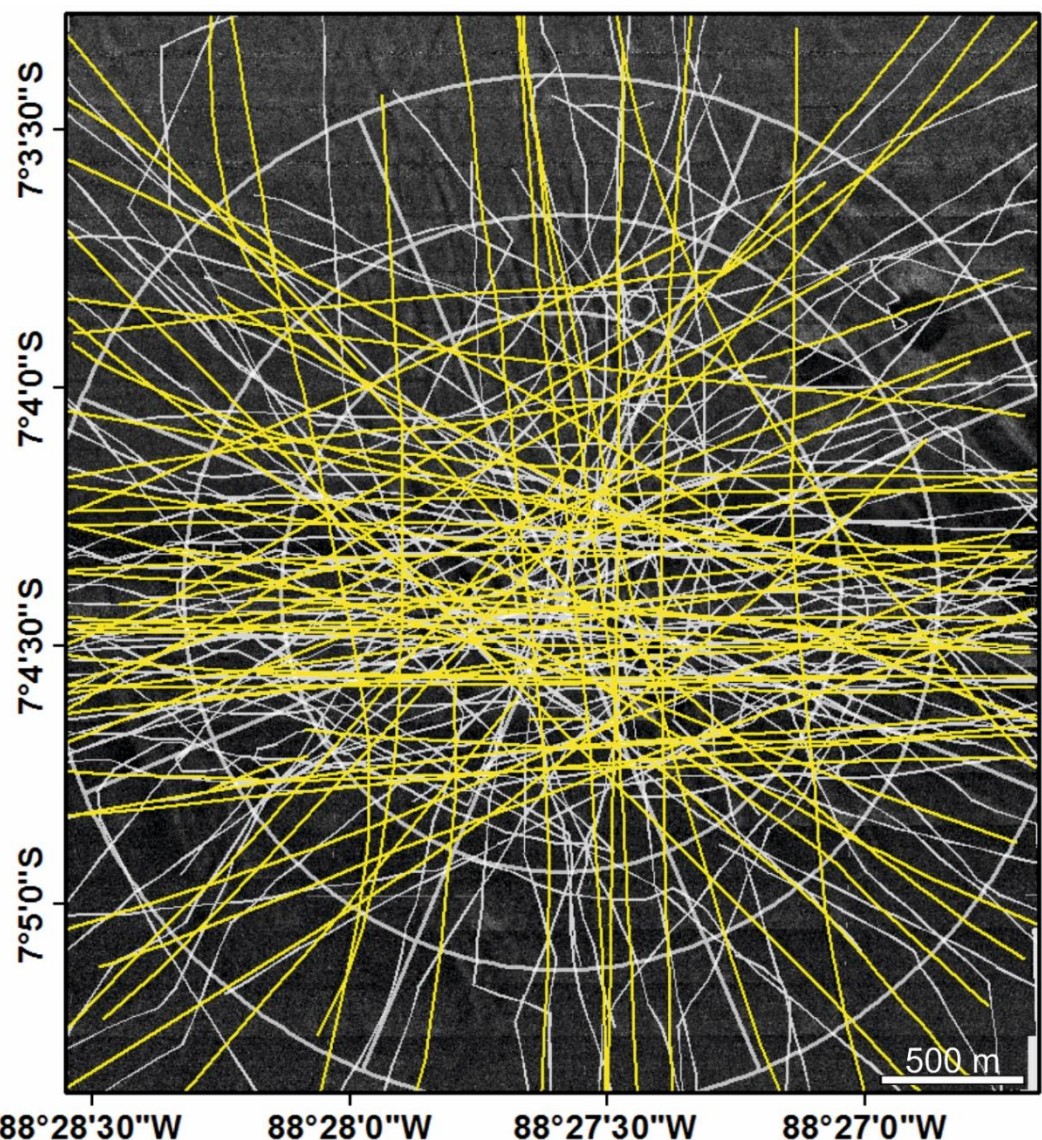

**Figure 10: Logged in 1989 (light grey) and reconstructed (yellow) plough track positions (Thiel and Schriever, 1989) plotted on the updated georeferenced AUV SSS map.**

### 3.3.2 Secondary sediment deposition impact

The derived sediment disturbance map of the DEA (Fig. 11) indicates the highest levels of disturbance within the center (C-sectors in Fig. 11) of the DEA coinciding with high densities of plough tracks and in the easternmost peripheral (P-sectors in Fig. 11) sectors.

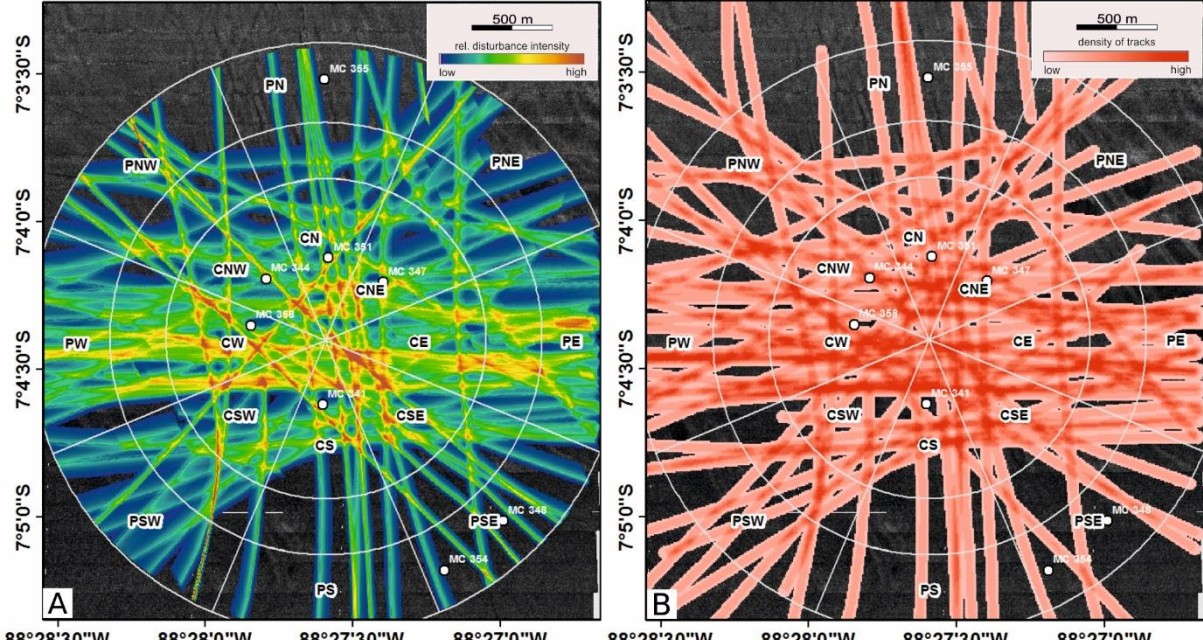

**Figure 11: Relative disturbance intensity and track-density maps of the DEA with the different sectors defined in 1989 (after Thiel and Schriever, 1989; "C"=Center; "P"=Peripheral). White dots indicate sampling stations from cruise SO77 (Schriever and Thiel, 1992) for XRAY sediment blanketing thickness analysis (results see Table H1); A) Relative disturbance intensity map of the DEA representing the disturbance by the track itself ("initial impact") and the thickness of re-settled plume sediments adjacent to the tracks ("secondary impact"). B): Heat map representing the track densities per 8x8m cell size indicating the relative plough disturbance levels within the DEA; the highest calculated density is set to 100 % disturbance.**

## 4 Discussion

### 4.1 Geo-referencing

The quality of geo-referencing different data layers towards each other highly depends on prominent morphological features that are detectable in all available data layers. The depth differences of more than 5 m between the ship-and AUV-based MB data after geo-referencing (red marked in Fig. E1) are related to two different AUV surveys, which seem to be inconsistent. However, the range of vertical depth deviation is still within the given depth-resolution of the EM122 ship system (Kongsberg, 2007) of max. 0.2 % of the water depth (= ~8m).

The alignment of the SSS and optical data onto the AUV MBES data was reasonable easy due to the presence of the plough marks visible in all of the high resolution data sets; this provided a sufficient number of anchor-points over the entire area. A considerable shift of 80 m encountered during the geo-referencing highlights the importance of accurate navigation and being aware of these kinds of offsets and deviations between the different data sets. When detailed sampling is requested, especially in terms of e.g. small scaled (tens of meters) habitat mapping, wrong navigation and geo-referencing will make analyses and correct interpretations not possible.

The remaining shift of 14 m between the USBL recorded sample positions and the position visible in the photo mosaics (Figure 9) is, for the given purpose, within an acceptable range. Regarding the punctual position of

sampling compared to the lateral grid resolution of 38 m of the ship-obtained MBES data as base of the geo-referencing such an offset might be unavoidable without considerable more technical effort of seafloor based navigation systems that build a temporary reference system for all deployed gear. In our case the USBL transponder for each sampling gear was mounted 100 m (MUC) / 50 m (BC) above the gear on the wire which might be already one source of error, if the rope most likely is not vertical above the sampling gear while this touches the bottom. Small, not completely correct evaluated static offsets between GPS and USBL antenna as well as misalignments between motion reference unit and USBL antenna might cause another problem that is difficult to evaluate without dedicated tests. Therefore the remaining range of 10 to 20 m deviation should be considered 'normal' for sample interpretation and navigational accuracy between differently derived map and location data sets.

**4.2 Plough tracks and age succession**

About 77 % of the reported disturbance tracks (60 out of 78) could be identified, most of them based on the SSS data (section 2.4). The 24 track segments of class P might account for the missing 18 tracks (e.g. P04 has been assigned to D21, Table 1).

The high-resolution MBES data did not fully capture the disturbance tracks due to the small morphological differences between plough-tracks and the surrounding seafloor (circa 15-30 cm; Boetius, 2015) and the internal structure of the plough-marks. The reconstruction of the initial disturbance was mainly based on the SSS mapping because of the higher along-swath resolution of the SSS compared to the MBES data. The penetration depth of the plough-harrow in combination with its very characteristic pattern facilitates the detection of the disturbance tracks. Morphological changes that are ensonified perpendicular (tracks parallel to the AUV flight path) cause higher reflections of the emitted signal compared to perpendicular tracks for which the small ridges and valleys of a plough track are ensonified parallel (Lurton, 2017; Beunaiche, 2017). Thus some tracks can be seen more clearly in the SSS data and others, which also causes that the sequence at some crossings could not be finally determined. The very first disturbance tracks are clearly visible within the SSS data, again indicating that the amplitude of the signal reflectance cannot be used as an indicator for their relative age. This becomes even more evident when comparing acoustic and optical data of the AUV. Some tracks that were barely visible in the SSS image (resolution: 0.5 m) could be clearly detected in seafloor photographs. Following this, the most reliable data source to establish the relative age sequence is the image and video material recorded by the various devices (AUV, ROV, OFOS) deployed during SO242 and the OFOS data from the previous cruises. The different survey altitudes and operation plans influence the area that was covered by each instrument and the quality of the images (Greinert, 2015). The AUV photo mosaics turned out to show the best results in resolving the age relation of multiple tracks even in highly disturbed areas within the DEA. There were a total of 9 AUV, 18 ROV, and 57 OFOS surveys conducted within the DEA between 1989 and 2015. However, since the DEA was not entirely covered by visual investigations, it is possible that some tracks which were not detected by the SSS were also not seen with the optical devices.

In general, the age reconstruction was successful, where more than one dataset was available. The plough tracks could be reconstructed with the highest amount of certainty for the very first and second set of disturbance tracks (PFEG 2 and PFEG 3). The uncertainties within the sequence regarding the absolute ages especially with later

sets of tracks (PFEG 4-11) increase since they are mainly based on statistical information and logical method of elimination (see section 2.4).

**4.3 Disturbance levels in the DEA**

Meso-scale numerical sediment distribution modeling by Jankowski et al. (1996, 2001) considering all plough
tracks of the DEA experiment predicted blanketing of resettled material of >100 gm$^{-2}$ up to a distance of 2 km. Due to the lack of data that measured the amount of re-suspended sediment, actual mass values of the blanketing cannot be given for our near field estimate approach. The settling velocity is also highly dependent on the sediment plume concentration (Gillard et al. 2019). As said, only 90% of the sediment is assumed to settle immediately due to flocculation and aggregation causing resettling of particles within proximal distances
(Becker et al., 2001, Gillard et al. 2019). The changing current conditions over the course of the plough experiment, especially in the later phases of the disturbance with a clear semi-diurnal signal (Thiel and Schriever, 1989), combined with the residence time of the re-suspended particles in the water column for more than 10 hours (Thiel and Schriever, 1989; Greinert, 2015) indicate that these remaining 10% were most likely spread across the entire DEA and beyond. The sediment blanketing map should thus be considered as the
minimum impact, with the SE sector being least impacted as already suggested by Thiel and Schriever in 1989.

The sectors with the highest sediment blanketing are CSE and CW (Fig. 11A), where also a high density of disturbance tracks occurs (Fig. 11B). X-ray studies aiming at measuring the deposition thickness were performed on selected MUC samples during SO077 (Fig. 11); results imply that sectors CS, CN, PSE and CNE are most heavily influenced with thicknesses between 5 and 30 mm (Schriever and Thiel, 1992, Table H1). In the
disturbance map for example within sector PSE only low disturbance is indicated, due to the very low density of tracks. A sample taken in sector CW (SO077_110MC_358, Fig. 11, Table H1) only shows a thin re-sedimented layer (1-2 mm), despite it is located in one of the most heavily disturbed areas (Fig. 11B) with high blanketing (Fig. 11A). This discrepancy could be explained by the more inaccurate positioning during SO77 (positions represent the ships position at the bottom contact time of the sampling gear) which hinders a punctual
comparison on such small scale. As implied in the disturbance map in the close vicinity of sampling station MC_358 there is a ~ 100 x 70 m wide patch where only thin sediment blanketing has been calculated (Fig. 11A). In this respect differences between the ships logged position and the devices position on ground in 2015 varied sometimes by more than 100 m (Greinert, 2015). In 1992, this distance might have even been greater as the old RV SONNE did not have dynamic positioning systems. Considering the high blanketing thicknesses proximal to
the tracks a sample location offset of several tens of meters could considerably change the result. Samples during SO077 could have been taken within or next to a track or from one punctual location within a disturbed area, where not much sediment has been deposited (Fig. 11A). This again highlights the importance being aware of the exact sampling positions and thus the need for detailed geo-referencing for the interpretation of the data.

However the generation of the disturbance intensity map is based on simplifications, not considering the specific
sediment settling parameters as particle sizes, density of particles and water turbulence. It also didn't include the local morphology, which has been proven to influence the sediment plume distribution (Peukert et al., 2018). Furthermore the micro-relief of ripple-crests and furrows within the track will also have an influence on the sediment blanketing thickness results from sediment cores, which again requires detailed position knowledge for

accurate sample interpretation. These factors could also be a reason for the deviating results from the SO77 x-ray studies compared to the disturbance map of this study. For more detailed investigations this should be considered and implemented into calculations and further sampling methods to allow an appropriate comparison of the results.

### 4.4 Sediment cover evolution through time

The numerous optical data acquired by OFOS, AUV and ROV during all expeditions to the DEA facilitate a comparison of the impact and its evolution over the 26 years that passed between the first and the most recent visit to the DISCOL area. Due to the explained navigation uncertainties especially during the early visits to the DEA (SO061, SO064) a direct comparison of exactly the same square meter of the seafloor is difficult but the comparison of different locations within an about 150 m long section of one track seems more reasonable (Figure 12). Generally, the fine morphology of the disturbance tracks appears to be smoothed out over time by currents and natural sedimentation, although the characteristic sequence of alternating crests and valleys is still clearly visible after 26 years (Boetius, 2015; Greinert, 2015).

Track V02 is one of the first tracks that have been created during PFEG 2 (Table 1) and could still be detected in different OFOS surveys in 1989 (SO61_OFOS10), 1990 (SO64_OFOS19) and 2015 (SO242_1_OFOS05; Figure 12A). Figure 12B shows the track only a few hours after it has been created. The characteristic plough structures are very prominent and the freshly broke up sediment lumps appear brighter than the surrounding sediments. Half a year later, the track appears distinctly smoothed and covered by sediment (Figure 12C) due to the re-settled sediment from plough deployments PFEG 4 to 11. Using the assumed sediment plume distribution of 120/100 m down-current and 20/30 m up-current the location where OFOS 19 crossed track V02 the observations are within the proximal deposition areas of later PFEGs (Figure 12A). The high sedimentation visible within this track is in accordance with the predictions of the disturbance map (Fig. 11A). Over the following 25 years until 2015 (Figure 12D) the track structures continued to be smoothed out, but the differences between 1990 and 2015 are less distinct. This illustrates the immense impact of the evolving sediment plume and the proximal re-settling of the sediment compared to the natural sedimentation and current induced shaping of the seafloor in the deep sea.

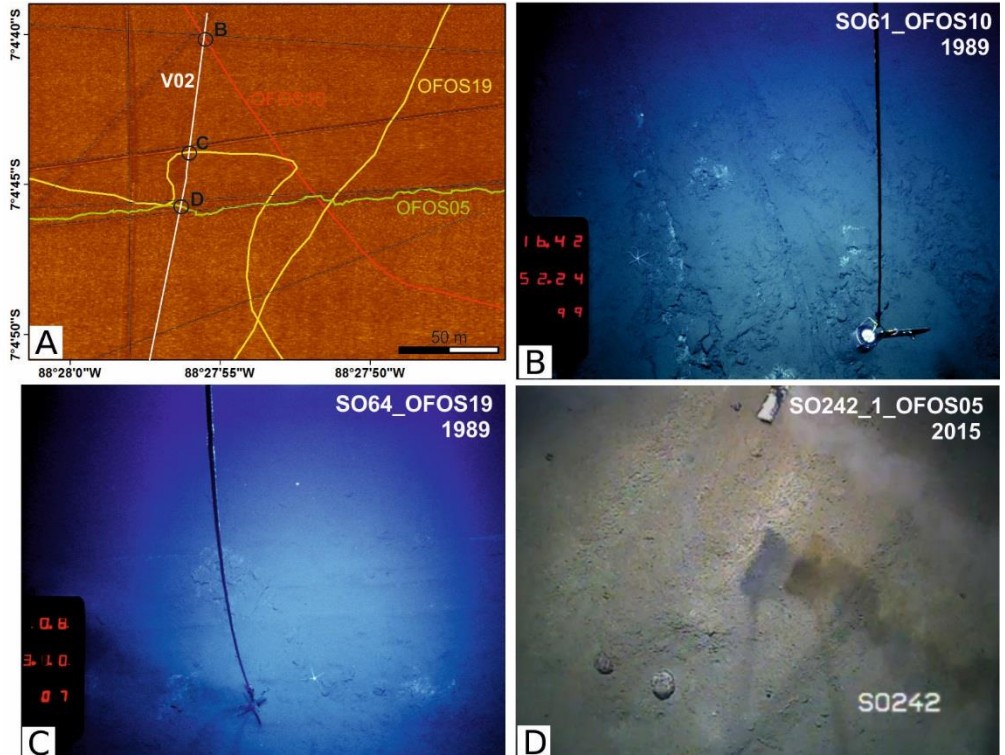

**Figure 12: Evolution of Track V02 over 26 years: A) red: SO61_OFOS10, yellow: SO64_OFOS19, green: SO242_1_OFOS05; white line indicates Track V02; black lines indicate surrounding Plough marks (younger than V02). B) Plough track a few hours after creation; C) After 6 months distinct smoothing of the track structures; D) after 26 years the track marks are still clearly visible but appears with distinct smoothed ripples. The identification of V02 in the different surveys was based on the orientation of the track being located close to another vertical running track west of V02 (indicated by the clack line in A), which was also crossed by all OFOS surveys before. The interval of the vertical tracks and being the first two vertical running tracks coming from the East, the identification of track V02 was successful and clear.**

Track H15 was one of the last tracks, created during PFEG 11 (Table F1).This track could be captured in OFOS dives from different deployments in the center of the DEA; H15 was further covered by the AUV photo mosaic of SO242 (Figure 13A). Fig. 13B of SO61 (OFOS17 from 1989) shows the freshly ploughed sediment within the disturbance track comparable to Fig. 12B. During SO106 in 1996 the track morphology is smoothed but broken up sediment lumps are still visible (Fig. 13C). The sediment cover within this track appears less than for track V02. The smoothing continued until 2015 but the track ripple structures are still apparent. At one location captured in the photo mosaic the H15 track crosses the V02 track (Fig. 13A), which allows a direct comparison of two tracks from different PFEGs. In 1992 V02 appears already much less distinct (Fig. 13E) than track H15 four years later (Fig. 13C), again pointing at strong re-sedimentation initiated by the plough activities after PFEG2. In 2015 the track ripples appear even weaker for V02 (Fig. 13F). This illustrates that the still observable levels of the secondary disturbance through the sediment plume need to be interpreted with respect to their sequential age and respective PFEG deployments. It underlines the importance of carful interpretations of the disturbance state of samples inside and near tracks. Furthermore, the track orientation with regard to the bottom current direction plays an essential role in terms of estimating the ecological impact coming along with the sediment plume; V02 runs parallel to the prevailing current direction causing higher sedimentation within and

very close to the track, whereas H15 runs perpendicular to the north- or south-directed bottom currents, which transported the sediment plume away from the track.

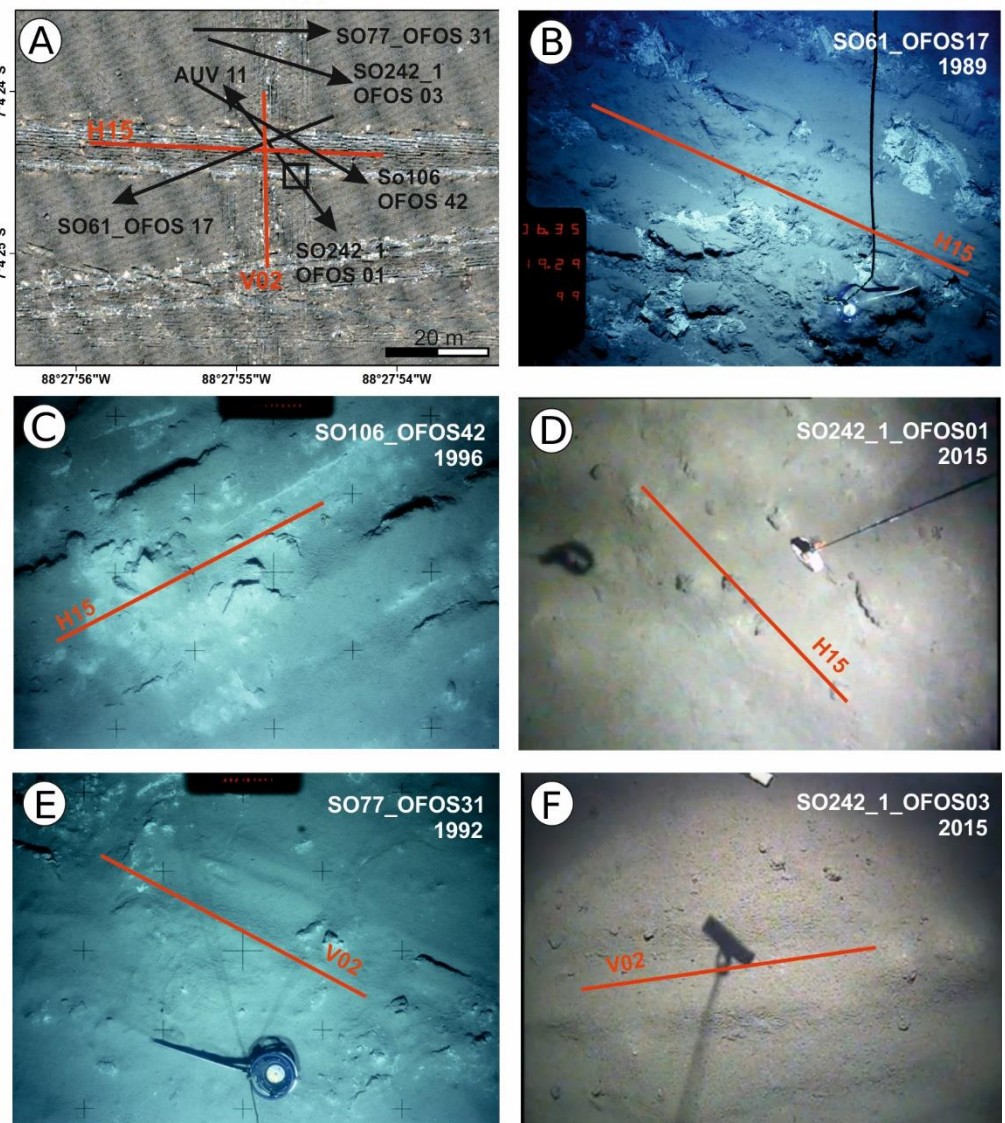

**Figure 13: Evolution of Track H15 and V02 over 26 years. A) AUV-acquired photo mosaic of a cross section in the central DEA, where OFOS Data from the different DISCOL visits could be**
**compared. Plough track H15 was one of the last tracks (PFEG 11) being created in 1989. B) The plough mark is characterized by cm-sized freshly broke up and shifted sediment piles, which appear distinctly smoothed in 1996 (C). 26 years after its creation the track is still apparent, but the structures appear evenly covered by natural sedimentation (D). E) Plough track V02 (PFEG 2) appears much more smoothed by higher re-sedimentation within the track in 1992, than H15. F)**
**After 26 years the characteristic ripple structures of the plough track appear only very weak.**

**Conclusions**

Results of our combination of legacy data from 1989 to 1996 with data from 2015 clearly indicate that underwater navigation and determining the accurate position of a seafloor sampling or observation location has been and still is difficult even using state-of-the-art technology. The common approach used in this study that

utilizes multiple hydroacoustic data sets of different resolution that are referenced against an absolute GPS-based data set (ships bathymetry) improved the overall accuracy. This is a pre-requisite for effective monitoring of deep sea impacts from deep sea mining or other spatial impact. Modern USBL and LBL systems linked with DVL and INS navigation on ROVs and AUVs can result in an absolute location accuracy of < 5m, which should be at. High resolution visual and acoustic data from AUV surveys emerged as a very resourceful tool for deep

sea surveys in general and monitoring impact experiments or even deep sea mining long-term effects in particular.

The re-georeferenced plough mark-positions and the estimated sediment plume distribution allow a more precise evaluation of the primary and secondary disturbance. With respect to uncertainties in under water navigation of up to hundreds meters this knowledge is essential for a correct interpretation of physical and optical samples.

The geo-referencing of all available optical data from the different cruises to the DEA allowed a quasi-direct comparison of individual tracks over a time span of 26 years. This gave a unique insight into the temporal change of the in-track morphology through blanketing. Results underline the creation of a strong plume-induced sedimentation compared to the normal sedimentation. This will cause harm to the low-sedimentation regime adapted deep sea ecosystem when industrial scale deep sea mining would occur. In this respect the results shown

here are not unconditionally comparable to the impact of such a large-scale and long-lasting operation (Gollner et al., 2017). The absolute deposition will be much more as the top 10 (or more) cm of the sediment will be suspended, gravity flows will most likely be generated. The amount of fine grained material remaining in the water column might be more as well and sediment blanketing most likely occurs up to tens of kilometers beyond the mined area (Boetius and Haeckel, 2017).

Detailed investigations are needed in coming impact experiments that should quantify the amount of sediment that is being re-suspended to enable a conclusive interpretation of the quantitative results for sediment blanketing analyses (be it through visual, sedimentological or chemical means). Knowing bottom currents and the local bathymetry in high spatial and temporal resolution are a fundamental pre-requisite for future impact experiments. Technologies exist and workflows are in place for conclusive assessments.

**Data availability**

The final referenced hydroacoustic maps and photo mosaics (as GeoTIFF), as well as the disturbance tracks (as GIS-readable shape-files) are available in the research data platform PANGEA (https://doi.pangaea.de/10.1594/PANGAEA.905616). Underwater photographs were georeferenced and uploaded to an annotation database (www.biigle.de) and can be accessed there. The raw photo data from the

AUV camera surveys in the DEA can be found in Greinert et al., 2017 (https://doi.org/10.1594/PANGAEA.882349).

**Acknowledgements**

We thank the captain and crew of RV SONNE SO242/1 for their cooperation and valuable contribution to a successful cruise. The work was funded by the German Federal Ministry of Education and Research through the

Mining Impact project (grant no. 03F0707A) of the Joint Programming Initiative of Healthy and Productive Seas

and Oceans (JPIOceans). Financial support was also provided by the EU project MIDAS (FP7, grant agreement no. 603418).We express our gratitude to the GEOMAR AUV team for their splendid support and professional attitude during the cruise. Gerd Schriever and the other anonymous referee are thanked for their valuable comments and contributions to the manuscript. This is publication #34 of the DeepSea Monitoring Group at
GEOMAR.

**Appendix**

**Appendix A: Large-scale Benthic Impact Experiments and associated sediment plume studies**

Table A1.1: Review of relevant large-scale Benthic Impact Experiments (BIEs) and collection of results of the sediment plume distribution studies 1978-1993.

| Experiment, Year | Area | Duration/ disturbed area | Amount of re-suspended and discharged sediment | Disturbance gear | Monitoring methods | Plume distribution extent | References |
|---|---|---|---|---|---|---|---|
| OMI (March-May 1978) | DOMES A, central CCZ; Test mining | 1) 15h (1d),(2) 54h (3d), 3) 33h (4d) | 4.3 m³/min | Suction dredge towed on skis 1) Hydraulic system, 2) Air lift system | Prior test surveys, Photos, BC's, sed-traps, current meters, nephelometers, CTD | Thickness: several tens of meters; 15-150 µg/l; 16km from mining site; 6-7 days after mining; measured 2m above seafloor (double the normal particle concentration); Model: re-sedimentation thickness <1mm beyond 400m off the disturbance for single track. Mining scenario: 160km away from the source. | Burns et al. 1980, Ozturgut et al., 1978. Lavelle, 1981 |
| OMA (10th Nov 1978) | DOMES C; Test Mining | 18h (1d) | Only Surface discharge measured: flow rate:80-95P/s; particle conc.: 3-9g/P; s: bulk density: 1.03g/cm³ | Collector device with multiple dredge heads | CTD, nephelometry, particulate sampling; light profiles | Focus on surface discharge plume; No Information about the benthic plume. | Ozturgut et al., 1980 |
| OMCO (Nov 1978) | Pacific (DOMES C, central CCZ) | No Information | Removed 4cm layer; app. 1.5m wide | Remote-controlled (self-propelled) Miner RCM | No information | No information | Welling, 1981, Khripounoff, 2006, Chung, 2009 |
| DISCOL (1989) | Peru Basin | 11km², 3.7km in diameter; 78times; | No information | Plough harrow (8m wide) | BC's, MUC's, OFOS, current- and turbidity (failed) measurements | Up to 30mm over DEA, plume was visible 6h after last deployment; Model: bed covered >100g/m² 1-2km from source | Thiel and Schriever, 1990, Jankowski et al., 1996 |
| BIE-II (1993) | Eastern CCZ | 49 tows in 19 days; 150 x 3000km area, 1.6h / tow cycle: 78.4h for 49 tows | 4328 m³ / ~5000 t sediment; 23.5t/h (wet) | DSSRS (operated with 0.5m/s) 5kg/s | sediment traps (2m above seafloor), current meters, transmissiometer, MUC's, BC's, SSSs | Plume did not travel far, settled quickly as fluid flow; Heaviest blanketing within 50m downstream off the disturbance. Distinct decrease of suspension between 50m (1094mg) and 300m (360 mg) downstream; Traps: max 1mm re-sedimentation thickness; Photos: 1-2cm in near disturbance thickness zone | Trueblood and Ozturgut, 1997 |

**Table A2.2: Review of relevant large-scale Benthic Impact Experiments (BIEs) and collection of results of the sediment plume distribution studies 1994-1997.**

| Experiment, Year | Area | Duration/ disturbed area | Amount of re-suspended and discharged sediment | Disturbance gear | Monitoring methods | Plume distribution extent | References |
|---|---|---|---|---|---|---|---|
| JET (1994) | Japanese area CCZ | 19 transects over experimental area: 2 parallel 2km long tow zones. 2 weeks; 20h 27min (Jones, 2000) | 3521t (dry sediment); discharge of 2475m3 --> ~2m3/min | DSSRS (hydraulic); discharge of slurry 5m above seafloor | Moorings with sediment traps (2m above seafloor) and current meters (~5m above seafloor), MUC's (before and after), seafloor photographs (after disturbance) | Kringing: 700 m downstream, 300m upstream; max thickness 2.6mm; Photos: 3km long and 2.5km wide; heavy blanketing (> 0.24mm) within 100m off track | Fukushima, 1995; Barnett and Suzuki, 1997.; Yamazaki and Kajitani, 1999 |
| IOM-BIE (1995) | Eastern CCZ | 2000 x 1500m; 14 tows (each 2.5km long) | different available information: 1800 / 128 m3 | DSSRS | Deep sea camera tows, sediment samples, moorings, CTD mounted on disturber | Studies focused on physical and chemical properties; only sparse information available (Kotlinski, 1998) | Tkatchenko et al., 1998; Kotlinski and Stoyanova, 1998 |
| INDEX (1997) | Central Indian ocean Basin | 9 days, 42h 14min operation time; 3000 x 200m; NOTE: several hours between the tows, 3-4 tows/day | 3592.8t; 6087m³; 2.4 m³/min | DSSRS | CTD, towed cameras, BC's, MUC's, Moorings 7m above seafloor (sediment traps, current meters, transmissiometer) | Survey 6-12 days after disturbance: no suspension in the water column; resettlement within few days; Sediments do not distribute vertically upwards; Re-sedimentation during disturbance: 150 mg/m²/day; immediately after (5-6 days): 95 mg/m²/day; Pre-disturbance: 48 mg/ m²/day; Sediment travelled for up to 150 m from the disturbance; finer particles may remain in the water column | Desa, 1997; Sharma, 1997, Sharma et al., 2001 |
| MMAJ (1997) | North Pacific, Markus-Wake Seamounts | - | - | Hydraulic Pick-up system | Moorings (currentmeters and sediment traps), Camera surveys | - | Yamada and Yamazaki, 1998; Yamazaki et al., 2001 |

**Appendix B: Hydroacoustic and optical data sets acquired during cruise SO242_1**

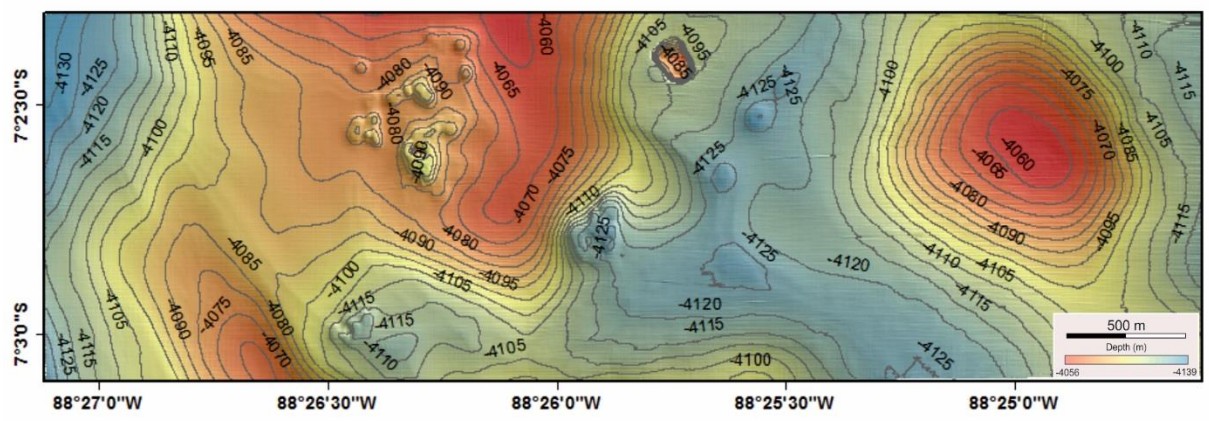


**Figure B1: AUV bathymetric data set from Abyss195-SO242/1_075-1 north of the merged AUV MB data from the other AUV MB missions during cruise SO242_1 (see Sect. 2.2). This data set has been shifted 9m down and 80m towards the East according to the ship-based MB data and the merged AUV MB dataset (see Sect. 2.3).**

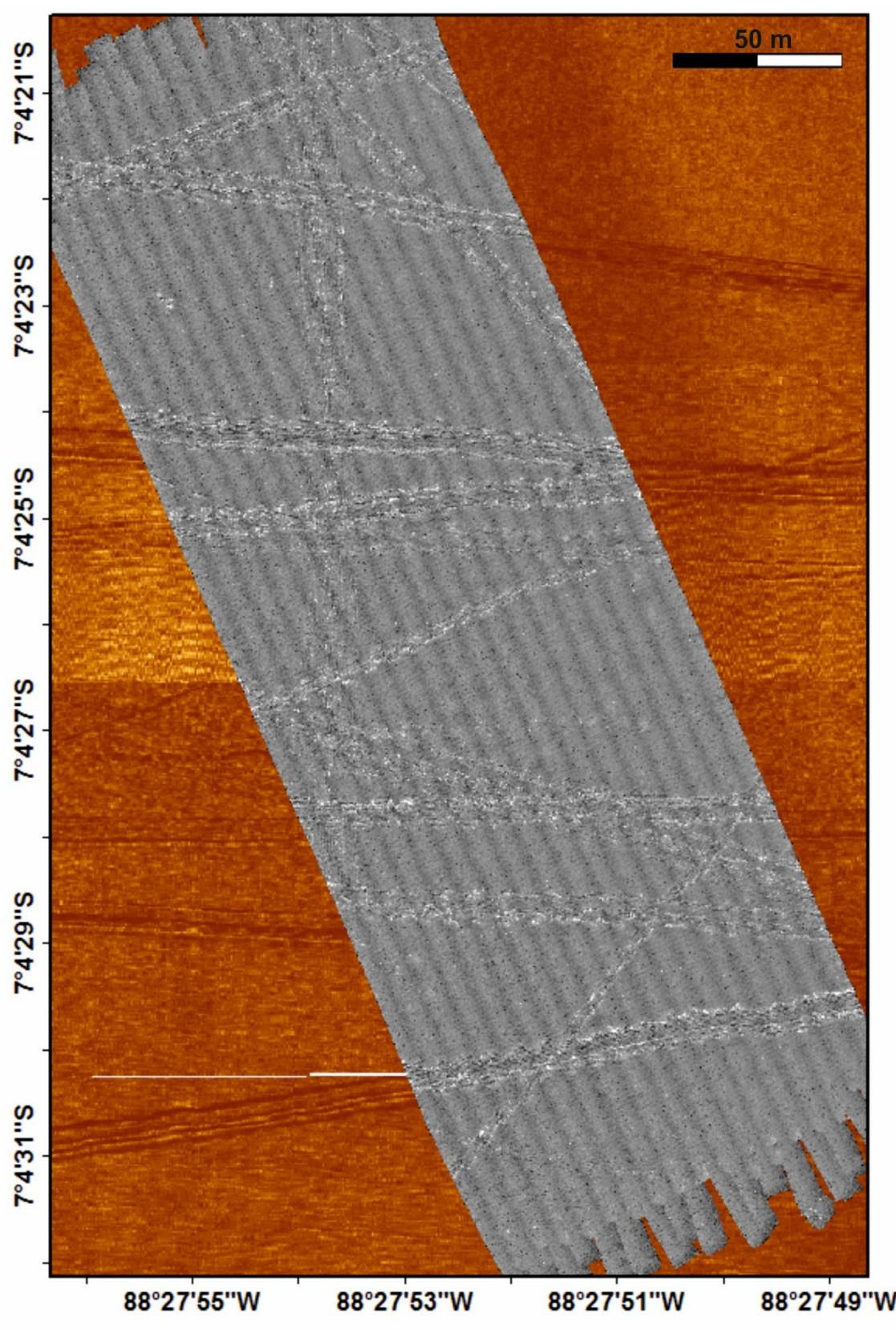


**Figure B2: AUC-acquired photo mosaic (Abyss199_SO242/1_102_1) within the central DEA (plotted on the SSS map) with a resolution of up to a few cm.**

**Terrain analysis of the working area / DEA**

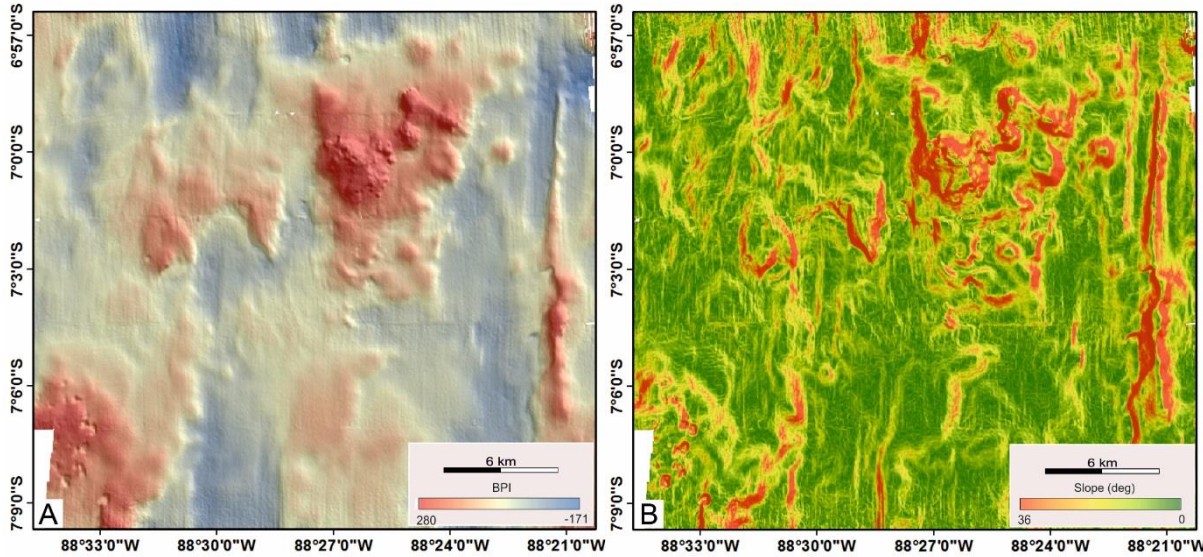

Figure B3: (A) BPI map of the ship-based bathymetric data indicating elevations (red colored) and basins (blue colored) within the area (scale factor 7600; grid cell size: 38 m, inner radius: 100 cells, outer radius: 200 cells). (B) Slope map of the ship-based bathymetric data indicating highest slopes in association with major morphological elevations. BPI and slope map have been calculated using the ArcGIS "Benthic Terrain Modeler (BTM)" add-in (Wright et al. 2012).

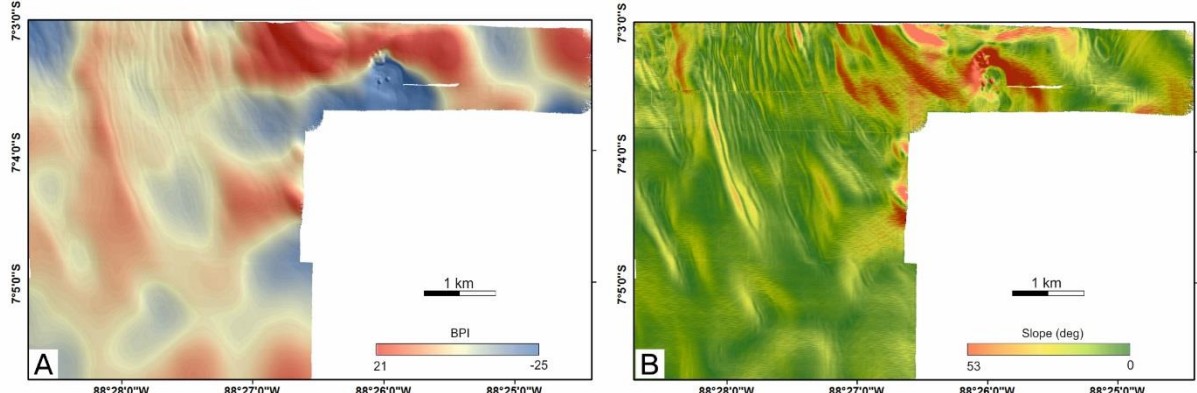

Figure B4: (A) BPI map of the AUV-acquired bathymetric data indicating elevations (red colored) and basins (blue colored) within the area (scale factor 1000; grid cell size: 2m, inner radius: 250 cells, outer radius: 500 cells). (B) Slope map of the AUV-acquired bathymetric data indicating low sloping terrain within the DEA and highest slopes NE of the DEA in the hilly terrain. BPI and slope map have been calculated using the ArcGIS "Benthic Terrain Modeler (BTM)" add-in (Wright et al. 2012).

**Appendix C: Alignment of the different available acoustic and optical data sets**

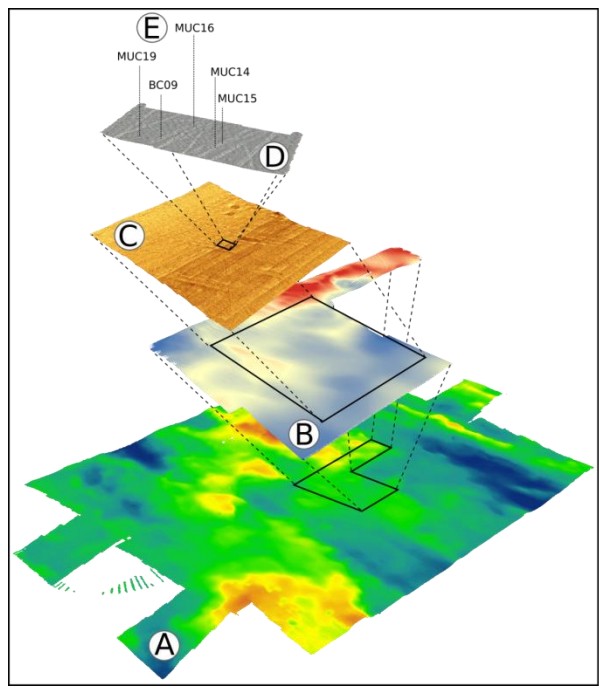


**FigureC1: Schematic representation and overview of the alignment steps of the different datasets: the ship-acquired bathymetric map (A) was set as base, where the AUV-acquired bathymetric data set (B) was aligned based on contour lines of prominent structures. The side-scan sonar map (C) and finally the photo mosaics created from seafloor images (D) were then fitted based on representative structures (see text and Figure 5 for details). The results were evaluated using the USBL positions of some seafloor sampling impacts visible in the images (E) as well as disturbance track sightings during ROV and OFOS dives during SO242-2 (not shown here).**


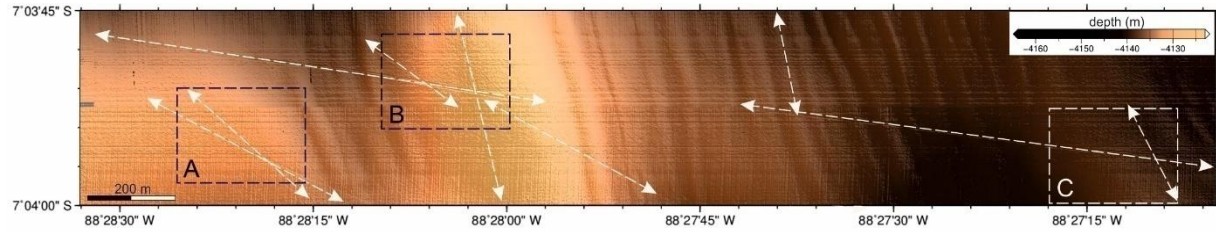

**Figure C2: AUV-acquired bathymetric map (1 m resolution) showing the context setting of the three inlet maps from Fig. 5A, B and C.**

**Appendix D: USBL positioning of sampling gear – bottom contact**

The sampling locations at the bottom were determined during the cruise using USBL navigation (Greinert, 2015); the USBL transponder was mounted on the cable approximately 50 m above the sampling gear (BC / MUC / GC) and the position was recorded every 7 seconds over the course of the entire operation. To determine the sampling location, the USBL data were edited and erroneous signals were removed before the data were

smoothed. The time of sampling and as a consequence the position was determined using the "wire tension" of the cable (Fig. D1).

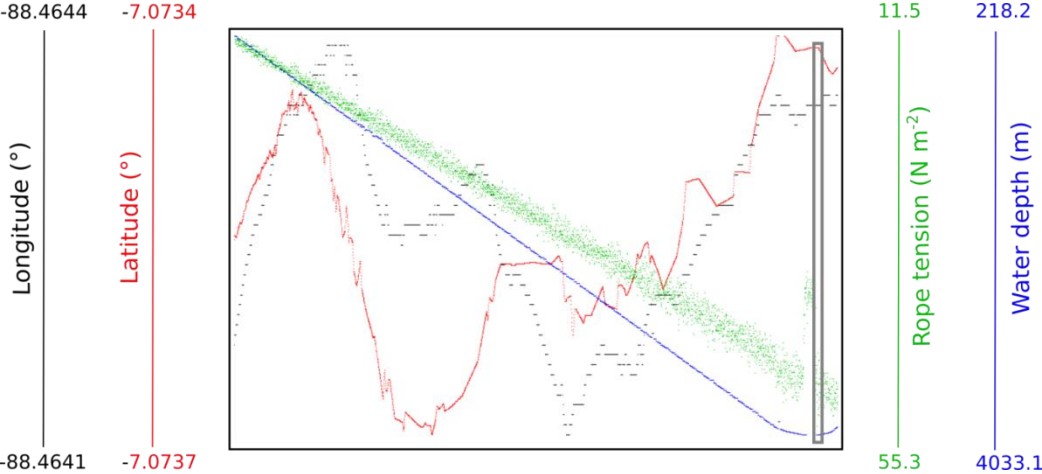

**Figure D1: Determining the seafloor sampling position using data from USBL communication with position (longitude (black) and latitude (red)) and water depth (blue). In addition, the rope tension (green) was also considered to improve the accuracy of the result. Once the sampling gear hit the seafloor, the tension of the cable suddenly dropped as a few meters of cable are still paid out so that the sampling gear is not unintentionally towed over the seafloor. The sampling location is**
**then identified as the position at the time when the "wire tension" increases again (black rectangle).**

USBL positioning has a minimum uncertainty of 0.2 % of the slant range at ideal conditions (iXBlue, 2016). Water depths in the DEA range between 4065.86 m and 4188.19 m, resulting in a possible error between 8.1 m
and 8.4 m with regard to the sampling location. Through continuous measurements and editing the data afterwards, additional errors such as the transponder being mounted approximately 50 m above the seafloor (Greinert, 2015) were minimized.

**Appendix E: Comparison of ship- and AUV-obtained data after alignment**

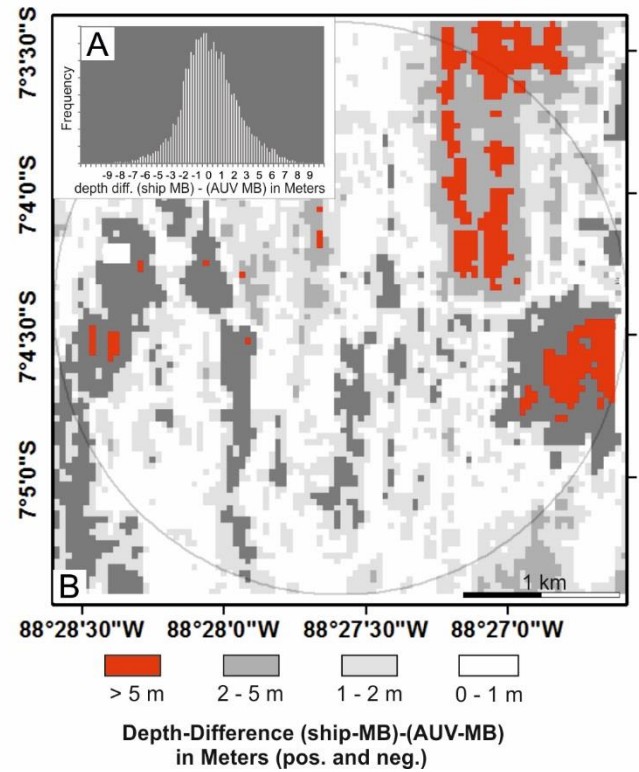

Figure E1: Grid (resolution 38 m) showing the differences in depth measurements between the ship- and AUV-obtained bathymetric data after the alignment of both data sets within the DEA. The values show a mean of 0 m ranging with a median of -0.5 m (A). Only in a few areas the deviation between the bathymetric datasets exceeds 5 m (B), possibly related to internal inconsistency between the different AUV MB surveys, which have been merged (see Sect. 2.2.1).



**Appendix F: Age sequencing of plough tracks**

**Table F1: Section from the logical 84x84 Matrix where all crossings with their relative sequence (older =earlier created; younger=later created; Matrix should be read column wise) were included. The numbers indicate the source of information, which the sequencing is based on: 1: only SSS; 2: nodule coverage-anomalies derived from AUV-photos; 3: AUV/OFOS images; 4: sequence derived from cross-referencing. ?: no intersection. The entire Matrix is not shown here due to its size but can be downloaded from the PANGEA data base (see "Data availability" https://doi.pangaea.de/10.1594/PANGAEA.905616).**

| Track_ID | V01 | V02 | V03 | V04 | V05 | V06 | V07 | ... |
|----------|-----|-----|-----|-----|-----|-----|-----|-----|
| V01 | xxx | - | - | - | - | - | - | |
| V02 | - | xxx | - | - | - | - | - | |
| V03 | - | - | xxx | - | - | - | - | |
| V04 | - | - | - | xxx | - | - | - | |
| V05 | - | - | - | - | xxx | 4 older | - | |
| V06 | - | - | - | - | 4 younger | xxx | 4 older | |
| V07 | - | - | - | - | - | 4 younger | xxx | |
| V08 | - | - | - | - | - | 1 younger | 4 younger | |
| V09 | - | - | - | - | - | - | - | |
| V10 | - | - | - | - | - | 1 younger | - | |
| V11 | - | - | - | - | - | - | - | |
| H01 | 1 older | 1 older | - | 4 younger | 1 younger | 1 younger | 1 younger | |
| H02 | 4 younger | 4 older | - | 4 younger | 1 younger | 1 younger | 1 younger | |
| H03 | 1 younger | 4 older | - | 1 younger | 1 younger | 1 younger | 1 younger | |
| H04 | 1 older | 1 older | - | 1 older | 1 older | 1 older | 1 older | |
| H05 | 1 older | 1 older | - | 1 older | 1 older | 1 older | 1 older | |
| H06 | 1 younger | 1 older | - | 1 older | 1 younger | 1 younger | 1 younger | |
| H07 | - | - | - | - | 1 younger | 4 younger | 1 younger | |
| H08 | 1 older | 1 older | - | 1 older | 1 older | 1 older | 1 older | |
| H09 | 1 older | 1 older | - | 1 older | 4 older | 1 older | 1 older | |
| H10 | 1 older | 1 older | - | 1 older | 1 older | 1 older | 1 older | |
| H11 | 1 older | 3 older | - | 2 older | 1 older | 1 older | 1 older | |
| H12 | 1 older | 3 older | - | 2 older | 1 older | 1 older | 1 older | |
| H13 | 1 older | 1 older | - | 2 older | 1 older | 1 older | 1 older | |
| H14 | 1 older | 3 older | - | 2 older | 1 older | - | - | |
| H15 | 1 older | 3 older | - | 2 older | 1 older | 1 older | 1 older | |
| H16 | 1 older | 4 older | - | 2 younger | 4 younger | 4 younger | 4 younger | |
| H17 | 1 older | 3 older | - | 2 older | 1 older | 1 older | 1 older | |
| H18 | 1 older | 3 older | - | 2 older | 1 older | 1 older | 1 older | |
| H19 | 1 older | 3 older | - | 2 older | 1 older | 1 older | 1 older | |
| H20 | 1 older | 4 older | 4 older | 4 younger | 4 younger | 4 younger | 1 younger | |
| H21 | 1 younger | 4 older | 4 older | 1 younger | 1 younger | younger | 1 younger | |
| H22 | 1 younger | 1 older | 4 older | 1 younger | 1 younger | 4 younger | 1 younger | |
| H23 | 1 older | 1 older | 4 older | 1 older | 1 older | 1 older | 1 older | |
| ... | | | | | | | | |

    **Appendix G: Offset between before and after the alignment of the different data sets**

Table G1: Distance between the seafloor sampling positions visible in the georeferenced photomosaic and post-processed USBL data positions.

| Station SO242_1 | Distance (m) USBL - Mosaic |
|:---:|:---:|
| BC06 | 6 |
| BC07 | 10 |
| BC08 | 12 |
| BC09 | 14 |
| BC10 | 13 |
| MUC14 | 17 |
| MUC15 | 20 |
| MUC16 | 15 |
| MUC19 | 25 |

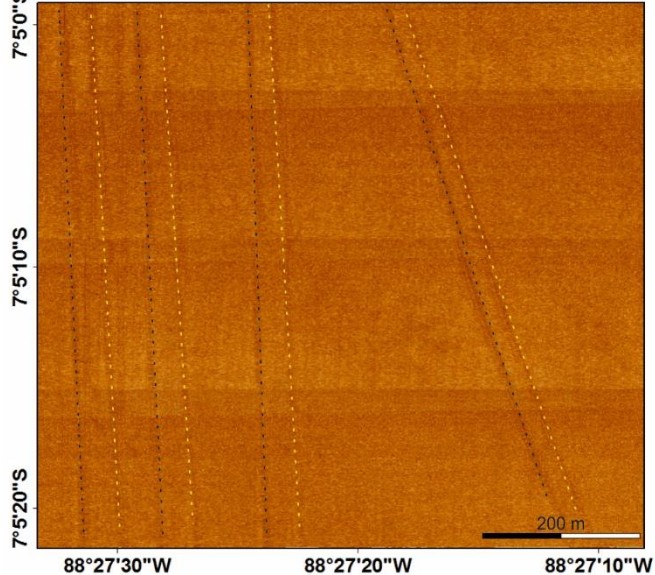

Figure G1: The plough marks visible in the SSS map show the offset of the SSS data before (yellow dotted) and after (black dotted) geo-referencing based on the MB data sets. An offset of approximately 30-50 m, in some areas also up to 80 m can be detected.

**Appendix H: Sampled sediment blanketing thicknesses from SO77 and SO242_1**

Table H1: Thickness of re-sedimented particles determined by X-ray analysis of samples within the DEA during cruise SO77 in 1992 (after Schriever and Thiel, 1992).

| Ship_Station | DISCOL Station | MUC | Sector | resedimentedlayer (in mm) |
|---|---|---|---|---|
| SO077_024 | 390 | 341 | CS | 10 - 30 |
| SO077_033 | 399 | 344 | CNW | 1 - 2 |

| SO077_055 | 418 | 347 | CNE | 7 - 10 |
|-----------|-----|-----|------|--------|
| SO077_056 | 419 | 348 | PSE* | 5 - 20 |
| SO077_075 | 439 | 351 | CN | 7 - 15 |
| SO077_092 | 455 | 355 | PN | 3 |
| SO077_110 | 471 | 358 | CW | 1 - 2 |

**Authors contribution**

FG has digitized the analogue available data from the initial DISCOL impact cruises, collected the entire set of
study-relevant data from all DISCOL cruises from 1989-2015 and did the post-processing of navigation data from SO242 deployments. He substantially contributed to the methodology and the document writing. AH contributed with literature review and to the methodology. She also created the Figures and was substantially involved in the document writing. TS and KK contributed with the acquisition, processing, curation and data analysis of optical data. KK created the mosaics of the AUV-acquired photographs. JG was the supervisor and
the initiator of this study, developed the study design and contributed to methodology strategies He also substantially contributed to the document writing.

**Competing Interests**

The authors declare that they have no conflict of interest.

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
