# Peer review of "Scars in the Abyss: Reconstructing sequence, location and temporal change of the 78 plough tracks of the 1989 DISCOL deep sea disturbance experiment in the Peru Basin"

_Biogeosciences, 2019_

## Referee Comment (RC1) · Anonymous Referee #1 · 21 Nov 2019

Review of manuscript bg-2019-348

**Scars in the Abyss: Reconstructing sequence, location and temporal change of the 78 plough tracks of the 1989 DISCOL deep sea disturbance experiment in the Peru Basin**

By

Gausepohl, Florian, Hennke, Anne, Schoening, Timm, Köser, Kevin,Greinert, Jens

General comments

This manuscript provides legacy data from previous cruises and new data from a recent research cruise from the Disturbance and Recolonization Experiment area (DISCOL) in the Peru Basin, SE Pacific. In 1989 an area of about 11 km² was ploughed using a plough harrow to simulate Mn nodule mining in this area. The data used in the provided study include ship-based multibeam bathymetric data (MBES), video data from deep-towed instruments as well as MBES, side-scan sonar and video data from an Autonomous Underwater Vehicle (AUV). The authors digitized and geo-referenced the old data, matched different data types (bathymetric, side-scan sonar and optical data) with different resolution and investigated the disturbance intensity of this area including sediment suspension and re-settling.

Major findings of this study are (1) old data with lower resolution and lower position accuracy can principally be used for comparison with modern high-resolution, high accuracy data provided a number of anchor points such as bathymetric features or sampling footprints are present, (2) there is an initial impact given through the mixing (ploughing) of the top 20 to 30 cm of the sediment and the related suspension of sediment into the bottom water (3) there is a secondary impact characterized by re-sedimentation of the initiated sediment plume and (4) the settling of the plume sediment is rapid in the immediate vicinity of the disturbance and causes high sedimentation rates which will be harmful to the benthic community which is not adapted to such high sedimentation rates.

Specific comments

The manuscript is well written and contains relevant references. Especially the methodology is well documented and convincing. However, I still have a number of issues the authors might take into account:

The description of data processing, i.e., how to match the old and new data, covers the largest part of the manuscript, whereas the discussion of the results and their implication (especially point (4) above) is rather short. A more in-depth discussion of the results is needed. Moreover, there are a number of repetitions mainly in chapters 1.3, 2.4 and 4.2 so that the manuscript should be shortened by removing these repetitions. This is already obvious in the abstract which mainly contains methods for data processing but no results!

The paper must critically review the fact that the DISCOL disturbance approach is very different to real nodule mining since no nodules were removed and sediments were only ploughed and not sucked into a device and subsequently dispersed a few meters above the seafloor as it would be done during real mining and as it has been done with the DSSRS. The manuscript does not say anything in this direction. Moreover, it should be discussed in this respect how the results of this study can be transferred to a real mining situation.

During reading I wondered about the significance of the age sequence of the disturbance tracks and why the authors put so much effort into it….It became clear to me in the lower part of the manuscript, i.e., to be able to differentiate between short-term settling of plume sediments with high sedimentation rates and natural sedimentation with low rates. Maybe it would be helpful if the authors present some clear objectives of their study within the introductory chapter.

As I already said above, this paper is mainly about the methods of data processing in order to compare old and new data with different quality. Some of these methodological approaches have been repeated a couple of times throughout the manuscript. I suggest that the authors should present a better separation of the method and the results of their study. In this respect they should provide a more in-depth interpretation of their results. For instance, they could discuss in detail the maps provided in figure 11.

I also suggest that the author might provide suggestions how precise navigation during Mn nodule mining impact studies should be and how this navigation accuracy could be realized, e.g. through the installation of a transponder array on the seafloor within which all instruments used on or above the seafloor should navigate.

Technical corrections

Apart from these comments, there are a number of special issues which I address below:

Line 36: References Kuhn et al., 2011 and Oebius et al., 2001 are missing in the reference list.

Line 59: Reference OMI; EC, 2013 is unclear and missing in the reference list.

Line 176: explain the abbreviation OFOS.

Line 208-209: What was the accuracy of the USBL system during the different cruises?

Line 224: the reference Devey et al. is missing in the reference list.

Line 225: ….between 4300m and 3850 m…

Lines 220 – 234: Please provide some information about slope angles.

Line 250 – 255 / Figure 3: How do the authors know that the NNW-SSE striking structures are ripple structures? To me they look like small grabens filled with sediment as well? On the east side of DEA these structures seem to be bent at their southern ends. Are these natural or artificial structures? Authors should discuss those obvious structures on the seafloor.

Figure 4: The contours shown in Fig. 4 are based on the ship-based MBES? Why? Why the authors didn't take the AUV-based MBES for the contours? If the latter is the case, please correct the figure caption.

Line 283 – 285: There is no information about the accuracy of the USBL sampling positions in sect. A2…The way how USBL position was detected is described in Appendix D, instead. But no information about accuracy is provided. Since USBL was probably run in transponder mode, accuracy is normally around ±0.2% of slant range, in this case, water depth (4000 m), i.e., accuracy should be ±8m. Is this correct?

Lines 286 – 294 (Fig. 5): What is the accuracy of the position of the sampling locations which act as anchor points?

Line 349: Were current measurements being carried out during the DISCOL experiments in 1989? Or how do the authors know the overall long-term current speed and direction?

Line 360-363: The capability of particles to flocculate is very important to consider (see also Guillard et al., 2019).

Line 400, formula 1: Provide reference for the application of this exponential function.

Line 470: Figure Caption of Fig. 11: Explain the abbreviations in the figure or give reference to where the reader can find this explanation.

Line 578: "….to the disturbance map of this study."

Line 610: There is no other N-S running track to the east of track V02 in Fig. 12A. But there are some other tracks running either E-W or ENE-WSW which were crossed by the OFOS stations during the different years.

Line 625-630: Is this conclusion supported by the disturbance intensity map presented in Fig. 11?

Lines 910 and 917: Reference Sharma & Nath, 1997 occurs twice.

Recommended reference:

B. Gillard et al. 2019. Physical and hydrodynamic properties of deep sea mining-generated, abyssal sediment plumes in the Clarion Clipperton Fracture Zone (eastern-central Pacific). Elem Sci Anth, 7: 5. DOI: https://doi.org/10.1525/elementa.343

---

## Referee Comment (RC2) · Anonymous Referee #2 · 13 Dec 2019

I think, the paper is a valuable contribution to the discussion about possible impacts of future manganese nodule mining and provides an overview about the activities since the end of the 70ies of the last century. According to my envolvement in this field I was able to comment on the first 30 years so my comments deal primarily with the description of the first 24 pages and some on the literature. These first 24 pages focus on the historical development of manganes nodule mining and possible impacts. Unfortunately the authors have mixed the nomencalture of what was done. So the activities in the late 70ies were pre-pilot mining test and the trial, to identify
possible impacts. In 1989 the first large scale impact experiment DISCOL was started in the South Pacific Ocean, focussing on the impacts on the bottom and sediment fauna, followed by the so called BIEs - (Benthic Impact Experiments) focussing on the impact of the resedimentation of sediments, transported in the water column during the mining process. DISCOl and the BIEs were not focussing on the same impact but complimentary to each other and the term benthic impact experiment should only be used for the the BIEs

Please also note the supplement to this comment:
https://www.biogeosciences-discuss.net/bg-2019-348/bg-2019-348-RC2-supplement.pdf

**Supplement:**

**Scars in the Abyss: Reconstructing sequence, location and temporal change of the 78 plough tracks of the 1989 DISCOL deep sea disturbance experiment in the Peru Basin**

5  Gausepohl, Florian[1,2], Hennke, Anne[1*], Schoening, Timm[1], Koeser, Kevin[1],Greinert, Jens[1,2]

[1] GEOMAR Helmholtz Centre for Ocean Research Kiel, Kiel, Germany

[2] Christian-Albrechts-University Kiel, Kiel, Germany

*Correspondence to:* Anne Hennke (ahennke@geomar.de)

10  **Abstract**

High-resolution optical and hydroacoustic seafloor data acquired in 2015 enabled the reconstruction of disturbance tracks of a past Benthic Impact Experiment that was conducted in 1989 in the Peru Basin in the course of former German environmental impact studies associated with manganese nodule mining. Based on this information, the disturbance level of the experiment regarding the plough impact and distribution and re-

15  deposition of sediment from the evolving sediment plume was assessed qualitatively. Through this, the evolution over the 26 years of a number of the total 78 disturbance tracks could be analyzed which highlights the considerable difference between natural sedimentation in the deep-sea and sedimentation of a resettled sediment plume. Such plumes are seen as one of the most concerning impact associated with potential Mn-nodule mining.

Problems in data processing became eminent while dealing with old data from the late 80s, at a time when GPS

20  was just invented and underwater navigation was in an infant stage. However, even today the uncertainties of underwater navigation and the use of a variety of acoustical and optical sensors at different resolutions require detailed post-processing in terms of absolute geographic positioning to improve the overall accuracy of the data. In this study, a ship-based bathymetric map of the survey area was used as absolute geographic reference and a workflow was applied successfully resulting in the most accurate geo-referenced dataset of the DISCOL

[revised manuscript text omitted]

155     processed high-resolution maps were used to guide sampling for box coring (BC), gravity coring (GC), multi-coring (MUC), as well as ROV and OFOS (Ocean Floor Observation System) dives that all were used for biological and geochemical studies to understand the level of disturbance, re-colonization and geochemical equilibration (Boetius, 2015; Greinert, 2015).

This study presents the best georeferenced data set of the study area through a combined processing of the

160     available ship- and AUV-obtained acoustic and optical data. In addition to this mapping exercise the succession of the disturbance tracks as well as their correct location is reconstructed, as this was not accurately documented in 1989. Although the 78 plough tracks were created over a period of two month (Thiel and Schriever, 1989) a more detailed understanding of their sequence is relevant regarding faunal differences from within or close to plough tracks in strong or weaker disturbed parts of the DEA as well as for understanding varying down-core

165     geochemical gradients that are effected by the thickness of the resettled sediment, the "blanketing" (Thiel, 2001; Boetius, 2015). Interpreting such biological or geochemical results correctly requires a very precise knowledge of the exact and absolute sample and footage location on the seafloor and their spatial relation to the tracks which are only a few meters wide and apart from each other. But even with today's highly developed positioning systems uncertainties and deviations of tens to a few hundreds of meters can occur when using different

170     underwater navigation systems and unavoidable different setups. This study further dealt with improving the even more inaccurate positioning of the legacy data from 1989 which is an indispensable task to conclusively define changes between 1989 and 2015 and spatial differences established already during the 
[revised manuscript text omitted]

**4.2 Plough tracks and age succession**

About 77 % of the reported disturbance tracks (60 out of 78) could be identified, most of them based on the SSS data. The 24 track segments of class P might account for the missing 18 tracks (e.g. P04 has been assigned to D21, Table 1). Some of the "P" tracks might be caused by the depressor that was attached to the cable in front of the plough-harrow during some of the disturbance phases (Thiel and Schriever, 1989).

Uncertainties during the reconstruction of the disturbance tracks sequence emerged as not all tracks show clear intersections with other tracks for the determination of the relative age. Combining multiple datasets uncertainties were mitigated to produce a final age sequence of the plough track to the best of our knowledge using all available information.

The high-resolution MBES data did not fully capture the disturbance tracks due to the small morphological differences between plough-tracks and the surrounding seafloor (circa 15-30 cm; Boetius, 2015) and the internal structure of the plough-marks. The reconstruction of the initial disturbance was mainly based on the SSS mapping because of the higher along-swath resolution of the SSS compared to the MBES data. The penetration depth of the plough-harrow in combination with its very characteristic pattern facilitates the detection of the disturbance tracks. Morphological changes that are ensonified perpendicular (tracks parallel to the AUV flight path) cause higher reflections of the emitted signal compared to perpendicular tracks for which the small ridges and valleys of a plough track are ensonified parallel (Lurton, 2017; Beunaiche, 2017). Thus some tracks can be seen more clearly in the SSS data and others, which also causes that the sequence at some crossings could not be finally determined. The very first 16 m wide disturbance tracks are clearly visible within the SSS data, again indicating that the amplitude of the signal reflectance cannot be used as an indicator for their relative age. This becomes even more evident when comparing acoustic and optical data of the AUV. Some tracks that were barely visible in the SSS image (resolution: 0.5 m) could be clearly detected in seafloor photographs. Following this, the most reliable data source to establish the relative age sequence is the image and video material recorded by the various devices (AUV, ROV, OFOS) deployed during SO242 and the OFOS data from the previous cruises. The different survey altitudes and operation plans influence the area that was covered by each instrument and the quality of the images (Greinert, 2015). The AUV photo mosaics turned out to show the best results in resolving

the age relation of multiple tracks even in highly disturbed areas within the DEA. There were a total of 9 AUV, 18 ROV, and 57 OFOS surveys conducted within the DEA between 1989 and 2015. However, since the DEA was not entirely covered by visual investigations, it is possible that some tracks which were not detected by the SSS were also not seen with the optical devices.

In general, the age reconstruction was successful, where more than one dataset was available. Combining all the different information an absolute sequence of the disturbance tracks could be reconstructed with the highest amount of certainty for the very first and second set of disturbance tracks (PFEG 2 and PFEG 3). Each of these PFEGs were followed by an OFOS dive. In these cases, the navigation data from 1989 were very useful as additional information about the ship's course at the time of the disturbance. 
[revised manuscript text omitted]

970

---

## Author Comment (AC1) · 21 Jan 2020

General comments

This manuscript provides legacy data from previous cruises and new data from a recent research cruise from the Disturbance and Recolonization Experiment area (DISCOL) in the Peru Basin, SE Pacific. In 1989 an area of about 11 km$^2$ was ploughed using a plough harrow to simulate Mn nodule mining in this area. The data used in the provided study include ship-based multibeam bathymetric data (MBES), video data

from deep-towed instruments as well as MBES, side-scan sonar and video data from an Autonomous Underwater Vehicle (AUV). The authors digitized and geo-referenced the old data, matched different data types (bathymetric, side-scan sonar and optical data) with different resolution and investigated the disturbance intensity of this area including sediment suspension and re-settling. Major findings of this study are (1) old data with lower resolution and lower position accuracy can principally be used for comparison with modern high-resolution, high accuracy data provided a number of anchor points such as bathymetric features or sampling footprints are present, (2) there is an initial impact given through the mixing (ploughing) of the top 20 to 30 cm of the sediment and the related suspension of sediment into the bottom water (3) there is a secondary impact characterized by re-sedimentation of the initiated sediment plume and (4) the settling of the plume sediment is rapid in the immediate vicinity of the disturbance and causes high sedimentation rates which will be harmful to the benthic community which is not adapted to such high sedimentation rates.

Author's response: The listing of the major findings to your understanding here is very valuable since it shows that some of the key findings and intentions of the paper need to be better emphasized. Major findings of this study are: (1) to present a most accurately georeferenced acoustical and optical data set of various data layers from the DEA including the best possible location of the plough marks created in 1989. The intention was to create one accurate data set which other scientist can use, now and also in 20 years time. (2) In this respect, the study also presents the sequence of origin of single or groups of plough marks. This is essential information for detailed interpretation of observations and particularly sediment samples for geochemical analyses as (3) the grade of disturbance in terms of thickness of resettled sediment differs distinctly between earlier and later created plough marks. (4) An additional objective was to follow the development of one track over 26 years with respect to re-settlement of organisms and sediment cover. Only accurate geo-referencing enables such direct comparison. (5) Finally we wanted to show that the impact of the resettled sediment plume just after/while the disturbance is much higher than the sedimentation over the

following 26 years. This is to point out that impact of a possible sediment plumes-inducing mining scenario is much higher than the low-sedimentation-rate-adapted deep sea environment is used to. We see the two major findings of point (2) and (3) listed of the reviewer more as fundamentals, which were named to define our terms used in this manuscript to distinguish between the two disturbance types. We hopefully managed to clarify and emphasize the key findings and objectives of the paper in the new version (see chapter 1.3).

Specific comments

Referee comment: The manuscript is well written and contains relevant references. Especially the methodology is well documented and convincing. However, I still have a number of issues the authors might take into account: The description of data processing, i.e., how to match the old and new data, covers the largest part of the manuscript, whereas the discussion of the results and their implication (especially point (4) above) is rather short. A more in-depth discussion of the results is needed. Moreover, there are a number of repetitions mainly in chapters 1.3, 2.4 and 4.2 so that the manuscript should be shortened by removing these repetitions. This is already obvious in the abstract which mainly contains methods for data processing but no results!

Author's response: The core content of this study is the presentation of the geo-referenced optical and acoustic data sets from the DISCOL area and the location of the plough marks created in 1989, since this information was lacking before that due to insufficient navigation accuracy back then. This is why the description of the data and the data processing take a large part of the manuscript. The presentation of the workflow and how data were acquired is supposed to be indicative for other studies dealing with the topic of environmental impact studies related to deep seafloor mining. It is shown that highly detailed information in respect of mapping and exact localization of sample positions is needed to facilitate a correct analysis of the data. We realized that the abstract was lacking in the presentation of the successful age sequencing of the plough tracks, which is also core content of the manuscript, since it leads to interpretations and results presented in sections 3.3, 4.3 and 4.4. Therefore in line 18 ff. this content was added. The results of the sediment blanketing studies are first applications to show the importance and the potential of this accurately geo-referenced data set for sample interpretations. Moreover in the mentioned chapters we do not see major repetitions. To our understanding the contents differ clearly in the different sections, summarized as followed: 1.3: Data acquisition during SO242; Motivation and content of study 2.4: Methodology of plough track identification and succession 4.2: Discussion of possible sources of error and evaluation of methods that best display the tracks themselves and best reflect the age succession (acoustic vs. Optical data) Nevertheless some repetitions within section 4.2 could be eliminated, since they are not essential to follow the text here and were mentioned before (in section 2.4), as the referee suggested: Line 509-514.: sentences have been deleted Lines 536 ff.: The plough tracks could be reconstructed with the highest amount of certainty for the very first and second set of disturbance tracks (PFEG 2 and PFEG 3). The uncertainties within the sequence regarding the absolute ages especially with later sets of tracks (PFEG 4-11) increase since they are mainly based on statistical information and logical method of elimination (see section 2.4).

Referee comment: The paper must critically review the fact that the DISCOL disturbance approach is very different to real nodule mining since no nodules were removed and sediments were only ploughed and not sucked into a device and subsequently dispersed a few meters above the seafloor as it would be done during real mining and as it has been done with the DSSRS. The manuscript does not say anything in this direction. Moreover, it should be discussed in this respect how the results of this study can be transferred to a real mining situation.

Author's response: This very important linkage to a possible real mining scenario was discussed in section 5 (Conclusions) Lines 655-616. It has been mentioned that this experimental setup is not unconditionally transferable to mining operations and that the amount of re-suspended material will be much higher, which in turn will influence

the behavior of the sediment plume. This exactly links to the problem that all existing sediment plume behavior estimations are assumptions since no "real scenario" experimental setups were possible in the past but would be necessary for significant predictions in this respect.

Referee comment: During reading I wondered about the significance of the age sequence of the disturbance tracks and why the authors put so much effort into it. . . .It became clear to me in the lower part of the manuscript, i.e., to be able to differentiate between short-term settling of plume sediments with high sedimentation rates and natural sedimentation with low rates. Maybe it would be helpful if the authors present some clear objectives of their study within the introductory chapter.

Author's response: We considered this valuable indication and inserted respective objectives within section 1.3.

Referee comment: As I already said above, this paper is mainly about the methods of data processing in order to compare old and new data with different quality. Some of these methodological approaches have been repeated a couple of times throughout the manuscript. I suggest that the authors should present a better separation of the method and the results of their study. In this respect they should provide a more in-depth interpretation of their results. For instance, they could discuss in detail the maps provided in figure 11.

Author's response: We tried to eliminate some of the repetitions through the manuscript (e.g. see specific comments #1 above). In some cases we find that some things need to be mentioned again to keep on track with the explanations. Lines 581-584 were shifted to section 2.2, since we agree here with the referee that this rather should be mentioned in the methods part (section 2). To our understanding the maps in Figure 11 are sufficiently discussed in section 4.3. For more detailed discussion we think more complex calculations are needed including more influencing parameters and also more detailed sample analysis would be essential for a more detailed verification of the

maps (discussed in section 4.3). As said in the manuscript, the disturbance intensity has been calculated based on assumptions and simplifications (section 2.5) to indicate trends and to give an idea about the distribution of the sediments that were re-suspended during the DISCOL experiment.

Referee comment: I also suggest that the author might provide suggestions how precise navigation during Mn nodule mining impact studies should be and how this navigation accuracy could be realized, e.g. through the installation of a transponder array on the seafloor within which all instruments used on or above the seafloor should navigate.

Author's response: As good as ever possible! Modern USBL and LBL systems linked with DVL and INS navigation on ROVs and AUVs can result in an absolute location accuracy of < 5m. This should be the aim, . . . and better if possible. We added this to the conclusions section in line 646.

Technical corrections

Referee comment: Apart from these comments, there are a number of special issues which I address below: Line 36: References Kuhn et al., 2011 and Oebius et al., 2001 are missing in the reference list.

Authors Response: The missing references have been added to the Reference list.

Referee comment: Line 59: Reference OMI; EC, 2013 is unclear and missing in the reference list.

Authors Response: EC,2013 is in the wrong place here and has been removed. OMI is the abbreviation for the named company Ocean Mining Inc.

Referee comment: Line 176: explain the abbreviation OFOS.

Authors Response: The abbreviation explanation has been added.

Referee comment: Line 208-209: What was the accuracy of the USBL system during the different cruises?

Authors Response: 1989 - SO061 (Thiel and Schriever, 1989): USBL (RS904) in conjunction with GPS. The GPS position varied considerably with navigation errors >55m. The USBL system was used when no GPS coverage was available. The position was then manually calculated relatively to the pre-determined center of the DEA resulting in a mean estimated standard error of less than 100 meters. 1989 - SO064 (Schriever, 1990): Acoustic transponder navigation system (SONATRACK III) in conjunction with AMF ATNAV transponders. Four transponders were deployed to get a reliable position fix. After an initial calibration, a relative geodetic system was established for DEA with a standard error of approximately 18.5 meters within the array. However, additional measurements using GPS and the associated SONATRACK positions revealed an offset between the relative and absolute geodetic system by 151 meters in 345 degrees true north and all positions had to be corrected for this error. After good initial results, the SONATRACK system failed and could not be used for the major part of the cruise, so that most of the underwater positioning was again recorded using the RS 904 system. 1992 - SO077 (Schriever and Thiel, 1992): No information is given regarding the underwater positioning of the then used OFOS/EXPLOS system. 1996 - SO106 (Schriever et al., 1996): No information is given regarding the underwater positioning of the then used OFOS/EXPLOS system, but since it was basically the same system as in 1992, most likely a similar setup was also used to determine underwater positioning during this cruise. The recorded signals had to be edited and smoothed manually to minimize the impact of faulty signals.

Referee comment: Line 224: the reference Devey et al. is missing in the reference list.

Authors Response: The missing reference has been added.

Referee comment: Line 225: . . ..between 4300m and 3850 m. . .

Authors Response: This has been corrected.

Referee comment: Lines 220 – 234: Please provide some information about slope angles.

Authors Response: The reasonable requested information about slope angles within the working area was implemented in line 223, 226, 234.

Referee comment: Line 250 – 255 / Figure 3: How do the authors know that the NNW-SSE striking structures are ripple structures? To me they look like small grabens filled with sediment as well? On the east side of DEA these structures seem to be bent at their southern ends. Are these natural or artificial structures? Authors should discuss those obvious structures on the seafloor.

Authors Response: The origin of these features is not clarified and was descriptively named ripple structures because of their appearance. These 'ripples' follow the local morphology parallel to the broader slope and terminate in depressions. We downloaded Parasound data from Boetius and Roessler, 2015 from PAGAEA and analyzed one profile that more or less perpendicularly crossed these 'ripples'. There are no indications for fault structures down to ~70m sediment depth that run parallel with the ripple-orientation (Figure 1); however the resolution of the ship-based Parasound data was not high enough to resolve the individual 'ripples' (opening angle=4°-4.5° => footprint of ~286m). Figure 1: Section of a sub-bottom profile (below) crossing some of the channel structures within the DEA (above). Red arrows indicte the crossing of such a structure. Because of no detected indications of faults in the shallow sub-seafloor and the wave-like undulations, which are not typically observed with faults we postulate that the ripples are not caused by faults or tectonic activity but are derived from sedimentary processes. The orientation of the structures follows the predominant NNW current direction in this area. We assume that bottom currents are channelized through the local trough around the rising terrain towards the NE and may cause turbulent flows which eventually cause furrowing. The side scan sonar data show that the 'ripples' are channel structures filled with softer sediment. The process of sediment furrowing has been described for the deep ocean seafloor where strong dominantly unidirectional bottom currents from 5-20 cm –s occur (Flood, 1983), which is the case here (see this manuscript, section 2.5. We took the advice of the referee and included a short discussion about the structures to the respective section (section 2.2, lines 255 ff.). We also changed the misleading term "ripple" to furrow channels" or short channels (line 255).

Referee comment: Figure 4: The contours shown in Fig. 4 are based on the ship-based MBES? Why? Why the authors didn't take the AUV-based MBES for the contours? If the latter is the case, please correct the figure caption.

Authors Response: The plotted contour lines are the ones originated from the ship-based MBES to illustrate the accordance of both MBES data sets after the geo-referencing and depth correction that was applied (lines 270-278). To clarify this we added a sentence for the explanation (lines 279-281).

Referee comment: Line 283 – 285: There is no information about the accuracy of the USBL sampling positions in sect. A2. . .

Authors Response: The link "A2" in line 285 was erroneous and has been corrected to "Appendix D".

Referee comment: The way how USBL position was detected is described in Appendix D, instead. But no information about accuracy is provided. Since USBL was probably run in transponder mode, accuracy is normally around $\pm 0.2\%$ of slant range, in this case, water depth (4000 m), i.e., accuracy should be $\pm 8$m. Is this correct?

Authors Response: Exactly this is discussed in Appendix section D, lines 745-747.

Referee comment: Lines 286 – 294 (Fig. 5): What is the accuracy of the position of the sampling locations which act as anchor points?

Authors Response: The USBL accuracy of the sampling positions has been discussed in Section Appendix D. The sampling positions visible in the photo mosaics georeferenced based on the hydroacoustic layers (lines 271-285) show a mean difference of 14 m (line 495) in comparison with the USBL positions, which is considered as acceptable range of deviation in this water depth.

Referee comment: Line 349: Were current measurements being carried out during the DISCOL experiments in 1989? Or how do the authors know the overall long-term current speed and direction?

Authors Response: In 1989 long-term current measurements were carried out in the DISCOL area in February and in March, documented in Thiel&Schriever, 1989. The results are reported in lines 375-384.

Referee comment: Line 360-363: The capability of particles to flocculate is very important to consider (see also Guillard et al., 2019).

Authors Response: We considered and indicated flocculation of the plume particles within the assumptions made for the disturbance intensity calculation, as mentioned in lines 364 ff. in section 2.5. The mentioned literature of Guillard et al. (2019) presents latest results in the field of sediment plume behavior, which supports the statements in the manuscript and therefore has been cited in the respective section 4.3.

Referee comment: Line 400, formula 1: Provide reference for the application of this exponential function.

Authors Response: The exponential function used to generate the disturbance map of the DISCOL area was chosen to account for the size-dependent particles settling within the assumed distances, as stated in section 2.5. In line 398 ff. the function is shown and explained; it is a very simplified function but considering the numerous uncertainties which influence the sediment plume distribution we consider this approach as a sufficient way to indicate trends of sediment blanketing qualitatively. We are aware of that for the true assessment of the impact this equation is not accurate enough as it does not considers the specific sediment settling parameters as particle sizes, density of particles and water turbulence. A respective comment was added in the manuscript lines 573-574 in section 4.3. However, given the setup of the experiment and the observations from deep-sea photographs and videos we believe that the majority of the impact in this case occurred in close proximity to the disturbance tracks (see section

2.5); this has recently also been indicated by Guillard et al. (2019).

Referee comment: Line 470: Figure Caption of Fig. 11: Explain the abbreviations in the figure or give reference to where the reader can find this explanation.

Authors Response: The Figure caption refers to the literature where the different sectors were defined. The explanation of the indications "C" and "P" has been added as suggested.

Referee comment: Line 578: "….to the disturbance map of this study."

Authors Response: The suggestion has been implemented.

Referee comment: Line 610: There is no other N-S running track to the east of track V02 in Fig. 12A. But there are some other tracks running either E-W or ENE-WSW which were crossed by the OFOS stations during the different years.

Authors Response: Lines 614-615: Thank you for this indication. Indeed, there is no vertical track running east of V02. It should have been "west". The sentence was corrected and clarified. Track V02 was chosen for analysis due to the low occurrence of vertical tracks in a larger radius. The close neighborhood with another vertical track west of V02, which has been crossed by all OFOS-surveys as well, allows an exact determination of V02 within the video footage. The more frequent occurrence of horizontally running tracks in consideration of the poor navigation accuracy of the OFOS tracks especially during the first cruises to the DEA impedes a definite identification of one track in different OFOS surveys. This explanation is implemented in the Figure caption of Figure 12 (lines 613-617).

Referee comment: Line 625-630: Is this conclusion supported by the disturbance intensity map presented in Fig. 11?

Authors Response: The disturbance intensity map indicates high disturbance in and in close vicinity of the cross section, since the track itself is set as highest disturbance (initial impact) and adjacent to the tracks the disturbance is also very high because

of the resettled sediments ("secondary impact", see section 2.5). So within the cross section a difference between the two tracks is not visible within this map but it is very distinct within the images in Figure 13.

Referee comment: Lines 910 and 917: Reference Sharma & Nath, 1997 occurs twice.

Authors Response: The duplicate has been removed and the order according to the publication year has been adapted.

References: Boetius, A.: RV Sonne Fahrtbericht/cruise report SO242-2 [SO242/2]: JPI Oceans Ecological Aspects of Dep-Sea Mining, DISCOL revisited, Guayaquil-Guayaquil (Ecuador) 28.08-01.10.2015, GEOMAR Reports, 2015.

Boetius, Antje; Roessler, Sebastian (2015): Profile of sediment echo sounding during SONNE cruise SO242/2 (DISCOL) in the Peru Basin, South Pacific Ocean, with links to ParaSound. Alfred Wegener Institute, Helmholtz Centre for Polar and Marine Research, Bremerhaven, PANGAEA, https://doi.org/10.1594/PANGAEA.854122

Flood, R. D. (1983). Classification of sedimentary furrows and a model for furrows initiation and evolution. Geol. Soc. Am. Bull. 94, 630–639. doi: 10.1130/0016-7606(1983)94<630:COSFAA>2.0.CO;2

Gillard, B., Purkiani, K., Chatzievangelou, D., Vink, A., Iversen, M.H. and Thomsen, L., 2019. Physical and hydrodynamic properties of deep sea mining-generated, abyssal sediment plumes in the Clarion Clipperton Fracture Zone (eastern-central Pacific). Elem Sci Anth, 7(1), p.5. DOI: http://doi.org/10.1525/elementa.343

Schriever, G.: Cruise Report DISCOL 2, Sonne - Cruise 64, Berichte aus dem Zentrum für Meeres- und Klimaforschung der Universität Hamburg, Nr. 6, Institut für Hydrobiologie und Fischreiwissenschft, Hamburg, 1990.

Schriever, G. and Thiel, H.: Cruise Report DISCOL 3, Sonne - Cruise 77, Report No 2, Reihe E: Hydrobilogie und Fischereiwissenschaft, Zentrum für Meeres- und Klimaforschung, Hamburg, 1992.

Schriever, G., Koschinsky, A., Bluhm, H.: Cruise Report ATESEPP (Impact of potential technical interventions on the deep-sea ecosystem of the southeast Pacific off Peru), Report No 11, Reihe E: Hydrobiologie und Fischereiwissenschaft, Zentrum für Meeres- und Klimaforschung, Hamburg, 195p., 1996.

Thiel, H. and Schriever, G.: Cruise Report DISCOL1, Sonne - Cruise 61, Berichte anus dem Zentrum für Meeres- und Klimaforschung der Universität Hamburg, Nr. 3, Institut für Hydrobiologie und Fischereiwissenschaft, Hamburg, 1989.

Please also note the supplement to this comment:
https://www.biogeosciences-discuss.net/bg-2019-348/bg-2019-348-AC1-supplement.pdf

––––––––––––––––––––––––––

---

## Author Comment (AC2) · 21 Jan 2020

Line 12:

(1) BIE's were only the US, Japanese, IOM and Indian Experiments - DISCIOL was a Re-colonization experiment and differed in the impact of the disturbance on the benthic community and did not target the impact of the created plume. Because of financial constraints DISCOL was not able to focus on this important aspect. Both experiments complement each other but are not at all comparable in their results.

[Figure]

(2) We thank the reviewer for this valuable comment to clearly differ between the terms BIE and Recolonization Experiment, which is the right term for this DISCOL case, as the comment author points out. We changed the nomenclature here and in the other cases appearing in the manuscript.

(3) lines 11-13: High-resolution optical and hydroacoustic seafloor data acquired in 2015 enabled the reconstruction of disturbance tracks of a past deep- sea re-colonization experiment that was conducted in 1989 in the Peru Basin during a German environmental impact study associated with manganese nodule mining.

Line 38:

(1) Literature recommendation

(2) We have looked in the pointed out paper and implemented it in the citation list since it points out another kind of manganese nodules as hard substrate habitat, which should definitely be named here.

(3) Line 38: Besides the removal of the Mn nodules as an important hard-substrate habitat on the abyssal plains (Purser et al., 2016; Vanreusel et al., 2016, Thiel et al. 1993), the mining activities will completely re-work the top sediment layers and re-suspend large amounts of sediment into the water column.

Line 46:

(1) See comment 1.

(2) We changed the used term "BIE" to the correct one "Recolonization Experiment.

(3) To evaluate these effects on the environment, several benthic impact experiments (BIE) and one Recolonization Experiment, the German Research Project "Disturbance and Recolonization Experiment-DISCOL" (http://www.discol.de), have been conducted in the past within different large Mn-nodule areas, including the Peru Basin (Thiel and Schriever, 1989), the Central Equatorial Pacific (e.g. Burns, 1980; Fukushima, 1995)

or the Indian Ocean Basin (Desa, 1997).

Line 70:

(1) Results of a monitoring were presented by Hessler during the International deep sea biology Conference at Hamburg. He told about the uncertainties of the results because they had not control of the sampling area and if they really hit the impacted areas or how close they were to the tracks or even inside the tracks. (personal communication with Hessler).

(2) The Deep Sea biology conference in Hamburg was in 1985. During the recovery studies from 2004 from the research expedition "NODINAUT" (RV "L'Atalante", IFREMER) the disturbance track was located and samples were taken within and close to the tracks as well as in reference areas. The submersible "NAUTILE" was used for sampling the tracks to ensure the correct positioning of the in-track samples (Miljutin et al. 2011). Hence the results cited here are plausible to us.

Line 74:

(1) All these experiments and monitoring efforts were based on small scale disturbances - maximum length of mining tracks 1.5 km, maximum width 4.5m. Based on Hessler's presentation at the International deep sea biology conference the DISCOL project was developed as the first large scale experiment in the deep sea ever! See also Thiel et al. 1998, Environmental risks from large-scale ecological research in the deep-sea: a desk study. Contract No. MAS2-CT94-0086, Commission of the European Communities, Directorate General for Science, Research and Development, Brussels, 210pp. )

(2) It is true that the scale of the DISCOL experiment is unique and the other listed experiments are not comparable in scale. They were named "large-scale" benthic experiments to point out the comparably large gear that were used for the disturbances and to mark the difference to comparable small-scaled disturbances created by other

devices, such as dredges or fish trailing marks for example. In 2015 a small distur­bance experiment was conducted where the sediment plume was created by the tow of an epibenthic sledge (Peukert et al. 2018). To point out that we exclude these kind of minor experiments we used the term "large-scale BIE's". To highlight the size of the DISCOL experiment we changed the sentence in lines 74-75.

(3) lines 74-75: Chronologically the next and largest ever created disturbance was conducted in the DISCOL Experimental Area in the Peru Basin.

Line 87:

(1) Add large scale.

(2) We added as suggested.

(3) Line 87 ff.: Again north of the equator, the first large-scale benthic disturbance experiment in the eastern Clarion-Clipperton Fracture Zone (CCFZ) conducted by the United States was the Benthic Impact Experiment II (BIE-II) in 1993, using the "Deep Sea Sediment Resuspension System" (DSSRS) (Brockett and Richards, 1994; Tsu­rusaki, 1997) as disturbance tool (Trueblood and Ozturgut, 1997).

Line 112:

(1) The right name is Kotlinski and Stoyanova.

(2) The spelling of the names has been corrected.

(3) line 112 ff.: In 1995, the InterOceanMetal (IOM) Joint Organization conducted a benthic disturbance experiment (IOM-BIE) over an area of 2000 x 1500m also in the eastern CCFZ, once more using the DSSRS (Kotlinski and Stoyanova, 1999; Radziejewska, 2002).

Line 131:

(1) Please see comment on page 3 - the activities in the late 70's were pre-pilot mining

test, what means industries tested the developed miners or components of these.

(2) The comment has been considered and the text was adapted accordingly.

(3) Line 131 ff.: Reviewing the different large-scale BIEs and pilot mining tests conducted between the late 70's and late 90's it becomes obvious that the different experimental setups and the missing uniform definition of 'a' plume (grain size distribution, flocculation behavior, total mass per liter, settling velocity etc.) make it impossible to use the presented information for a meaningful predict of the behavior of a sediment plume created during a real deep sea mining operation (Peukert et al., 2018).

Line 162:

(1) 4 weeks only! from the end of February to 3rd week of March - please see Cruise report

(2) The time span has been corrected to 4 weeks. The period of two months includes the pre-baseline studies and the first impact studies right after the disturbance, but the disturbance phase itself lasted only 4 weeks, as the reviewer pointed out.

(3) line 162 ff.: Although the 78 plough tracks were created over a period of 4 weeks (Thiel and Schriever, 1989) a more detailed understanding of their sequence is relevant regarding faunal differences from within or close to plough tracks in strong or weaker disturbed parts of the DEA as well as for understanding varying down-core geochemical gradients that are effected by the thickness of the resettled sediment, the "blanketing" (Thiel, 2001; Boetius, 2015).

Line 179:

(1) Please add papers of Bluhm.

(2) Bluhm indeed provided valuable information here and has therefore been added to the citation list.

(3) lines 175-179: Until 2015, the location and path of the disturbance tracks as well

as the position of video and photo material of the past OFOS (Ocean Floor Obser-vation System) surveys only existed as a vast collection of analogue (i.e. cruise re-ports, printed large navigational charts, video cassettes and slide films) and some digital records (i.e. OFOS annotation files, sample analysis as text or EXCEL files, e.g. Bluhm, 1994, Bluhm and Thiel 1996, Thiel and Schriever 1989, Schriever 1990, Schriever and Thiel 1992, Schriever et al., 1996).

Line 207:

(1) cite Bluhm

(2) The missing literature source was added.

(3) lines 206-209: Additional visual investigations during all cruises to the DEA were conducted using the towed camera system OFOS either equipped with both a still and video camera (Bluhm and Thiel, 1996, Thiel and Schriever, 1989; Schriever, 1990; Schriever and Thiel, 1992; Schriever et al., 1996) or just a video camera, which was mounted on the frame of a sampling device (Boetius, 2015; Greinert, 2015).

Line 353:

(1) This is OK, the tests of the 16m wide plow harrow was done outside of the DISCOL Area! Three or 4 tracks were done only. Handling of the 16m wide plough harrow on board of the ship was extremely difficult and took too much time.

(2) There was a misunderstanding in the time when the width of the plough harrow was halved. Since the focus here is on the DISCOL Area the 16 m are not considered. The manuscript has been adjusted to that.

(3) Line 354-356: Each track was assumed to have a width of 8 m, not considering the possible handling problems with the plough-harrow (e.g. being towed only on the side, short loss of bottom contact ,Thiel and Schriever, 1989)

Line 385:

(1) These investigations were no BIEs - they were a kind of monitoring of the impact of the pre pilot mining tests.

(2) The passage was corrected and the correct term was implemented.

(3) With regards to other impact monitoring results from large-scale disturbances (e.g. Lavelle et al., 1981, Table A1) and the results of small-scale disturbance experiments conducted during SO239 (Martinez Arbizu and Haeckel, 2015; Peukert et al., 2018), SO242/1 (Greinert, 2015), and SO242/2 (Boetius, 2015), the maximum distance affected by sediment blanketing was assumed to be 120 m with, and 20 m against the current direction for the "strong" current regime.

Line 410:

(1) please delete

(2) The passage has been corrected.

(3) The final blanketing map was produced by adding all relative sediment thicknesses within each square meter of the DEA area using the blockmean command in GMT (argument –Ss to get the sum; Wessel et al., 2013) and producing an interpolated grid using the nearneighbor command.

Line 596:

(1) The DiSCOL Area was visited 6 months (SO64) or three years (SO77) after the impact - not 1 year!

(2) This has been corrected. The cruise SO64 took place in autumn 1989, not in 1990. The Figure 12C was adapted accordingly.

(3) Half a year later, the track is distinctly smoothed and covered by sediment (Figure 12C) due to the re-settled sediment from plough deployments PFEG 4 to 11.

References Miljutin, D. M., Miljutina, M. A., Arbizu, P. M. and Galéron, J.: Deep sea

nematode assemblage has not recovered 26 years after experimental mining of poly-metallic nodules (Clarion-Clipperton Fracture Zone, Tropical Eastern Pacific), Deep. Res. Part I Oceanogr. Res. Pap., 58(8), 885–897, doi:10.1016/j.dsr.2011.06.003, 2011.

Peukert, A., Schoening, T., Alevizos, E., Köser, K., Kwasnitschka, T., Greinert, J.: Understanding Mn-nodule distribution and evolution of related deep-sea mining impacts using AUV-based hydroacoustic data, Biogeosciences 15, 2525-2549, https://doi.org/10.5194/bg-15-2525-2018, 2018.

Please also note the supplement to this comment:
https://www.biogeosciences-discuss.net/bg-2019-348/bg-2019-348-AC2-supplement.pdf

―――――――――――――――――

---

## Author Response (AR2)

Dear Jack Middelburg,

thank you for your valuable comments on our manuscript. We considered your recommendations and implemented the following changes:

**RC:** Line 75: (OMA) a second seafloor....  **AC:** Done
**RC:** Line 104: ... to these data, camera...**AC:** Done
**RC** Line 142: meaningful prediction... **AC:** Done
**RC** Line 143-146: reformulate, unclear... **AC:** Done
**RC** Line 155: even is not needed... **AC:** Done
**RC** Line 159: ... and related varying and habitats... this is unclear... **AC:** Done
**RC** Line 167: location are reconstructed... **AC:** Done
**RC** Line 180: in 2015, uncertainties... **AC:** Done
**RC** Line 240 and further: do not reference forward, i.e. avoid citing unpublished material. Moreover, if really needed then do provide title, authors, etc. ... **AC:** Done / Lines 240, 243, 246: added reference Melchior, 2017. Line 261: Here, the reference of Devey et al. is necessary, since they studied particularly these formations. The manuscript to date is in review process. In the literature list the reference is included (including names, title, etc.)
**RC** Line 277: cm s-1... **AC:** Done
**RC** Line 287: relied on Doppler... **AC:** Done
**RC** Line 333: are typically shorter... **AC:** Done
**RC** Line 370: blanketing the course... ... **AC:** Done
**RC** Line 404: no information is available about..material, the impact... **AC:** Done
**RC** Line 405: between 1 (within the disturbance tracks) and 0.1, representing... **AC:** Done
**RC** Line 440: data sets... **AC:** Done
**RC** Line 481: In comparison, the ... ... **AC:** Done
**RC** Line 540: some crossings, could not be (or otherwise rewrite sentence). ... **AC:** Done
**RC** Line 581: MC_538, there.. ... **AC:** Done
**RC** Line 595: investigations, this... **AC:** Done
**RC** Line 641: careful... **AC:** Done
**RC** Line 664: ...which should be at... and then something is missing. Please rewrite... **AC:** Done

[revised manuscript text omitted]